# Teaching LLMs to Abstain via Fine-Grained Semantic Confidence Reward

## Abstract

Mitigating hallucinations in Large Language Models (LLMs) is critical for their reliable deployment. Existing methods typically fine-tune LLMs to abstain from answering questions beyond their knowledge scope. However, these methods often rely on coarse-grained signals to guide LLMs to abstain, such as overall confidence or uncertainty scores on multiple sampled answers, which may result in an imprecise awareness of the model's own knowledge boundaries. To this end, we propose a novel reinforcement learning framework built on **Fine-grained Semantic Confidence Reward (FISCORE)**, which guides LLMs to abstain via sample-specific confidence. Specifically, our method operates by sampling multiple candidate answers and conducting semantic clustering, then training the LLM to retain answers within high-confidence clusters and discard those within low-confidence ones, thereby promoting accurate post-hoc abstention. Additionally, we propose a new metric for evaluating the reliability of abstention fine-tuning tasks more comprehensively. Our method significantly enhances reliability in both in-domain and out-of-distribution benchmarks.

## 1 Introduction

Large language models (LLMs) have achieved remarkable success, demonstrating powerful capabilities in content generation, complex reasoning, and software development (Achiam et al., 2023; Grattafiori et al., 2024; Yang et al., 2024). Their widespread adoption has established them as essential tools in numerous applications. However, a critical vulnerability threatens their reliability: their propensity to hallucinate, fabricating plausible but false information when faced with questions beyond their knowledge (Zhang et al., 2023). Such hallucinations erode user trust and can propagate misinformation. Reliance on fabricated content in high-stakes domains such as medicine or law can have severe consequences. Therefore, mitigating hallucinations is essential for the safe and responsible deployment of LLMs.

A promising direction for mitigating hallucination is to fine-tune LLMs to abstain from answering questions that lie beyond their knowledge scope and answer those within it (Wen et al., 2025; Cheng et al., 2024). Supervised methods often categorize a training set into "known" and "unknown" questions based on the correctness or accuracy of the model's answers, then train the model to answer the former and abstain from the latter (Zhang et al., 2024; Cheng et al., 2024). To avoid reliance on ground-truth labels, other approaches leverage uncertainty metrics such as semantic entropy to partition the training set into known and unknown questions (Zhang et al., 2024; Tjandra et al., 2024; Xue et al., 2025). However, these methods rely on coarse, global signals, such as a single correctness label or an aggregated uncertainty score, to guide abstention decisions. These approaches don't account for the nuanced confidence levels of individual samples, resulting in an ambiguous decision boundary. Consequently, models may either excessively abstain from answerable questions or fail to abstain when facing low-confidence inputs, which prevents the model from balancing truthfulness and helpfulness.

To overcome these limitations, we propose **Fine-grained Semantic Confidence Reward (FISCORE, /ˈfɪs.kɔːr/)**, a novel reinforcement learning framework that replaces the conventional global uncertainty reward with a fine-grained, **per-sample** confidence reward. The core insight is that an answer's intrinsic confidence is reflected in its semantic consensus among all samples, and such confidence can be fully utilized for factual alignment. Although existing methods also sample

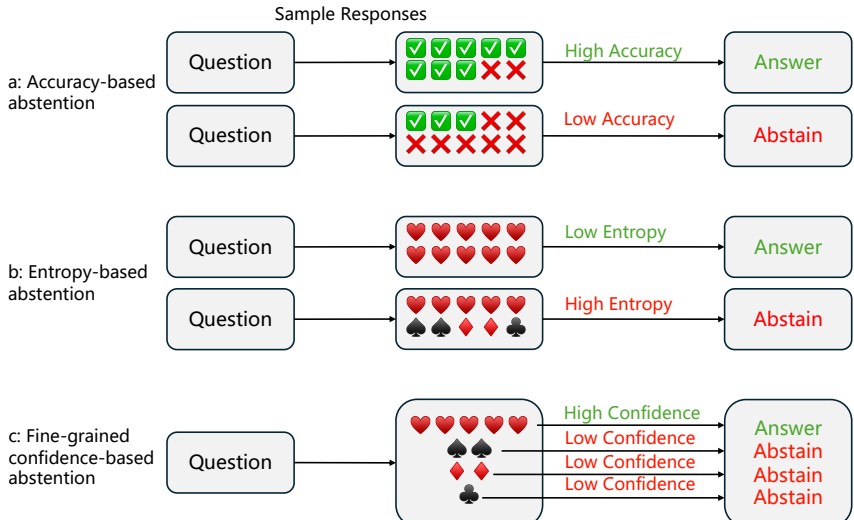

Figure 1: (a) Accuracy-based abstention. (b) Entropy-based abstention. (c) Ours: fine-grained confidence-based abstention.

multiple answers, they typically distill them into a single aggregated uncertainty score. In contrast, as shown in Figure 1, our approach rewards the model for accurately assessing the confidence of each sample. Specifically, we first generate multiple candidate responses and group them into clusters based on their semantic equivalence. We use the cluster size as a proxy for the intrinsic confidence of its member samples: a large cluster implies high confidence, whereas a small cluster implies low confidence. A positive reward is granted only when the model's expressed confidence (e.g., "sure" or "unsure") aligns correctly with this intrinsic confidence level. This fine-grained signal incentivizes the model to align its verbalized confidence with semantic consensus, thus enabling it to learn a more nuanced knowledge boundary and make more accurate abstention decisions.

Existing studies indicate that helpfulness and truthfulness are like two ends of a seesaw — enhancing one often leads to a decline in the other (Zhu et al., 2025). We notice that current metrics fail to accurately reflect both aspects, making them inadequate for assessing abstention methods. Regarding this, we propose a reliability metric grounded in the model's self-awareness of its knowledge boundaries. This metric synthesizes helpfulness and truthfulness, aiming to prevent models from over-rejecting queries or generating hallucinations.

Our main contributions are as follows:

- We design a fine-grained semantic confidence reward (see Section 3.3), which significantly improves the reliability of LLM.
- We propose a new metric (see Section 2.2) that integrates the helpfulness and truthfulness of LLM into a single reliability metric, offering a comprehensive assessment of its awareness of its own knowledge boundaries.
- We experiment with various LLMs and test them on both in-distribution and out-of-distribution data. The results demonstrate that our approach outperforms baselines in most cases, particularly on out-of-distribution data, verifying the effectiveness and generalization capability of the proposed fine-grained semantic confidence reward.

## 2 PROBLEM FORMULATION

### 2.1 ABSTENTION FINE-TUNING

To mitigate the risk of hallucination inherent in LLMs, a primary alignment strategy is to fine-tune models to selectively abstain from answering questions that fall beyond their knowledge boundaries (Wen et al., 2025; Li et al., 2025a). The objective is to produce a model that is both helpful, by

Table 1: Abstention confusion matrix.

| Refined LLM / Initial LLM | Correctly answered | Incorrectly answered | Abstained |
|---|---|---|---|
| Known Questions | $N_1$ | $N_2$ | $N_3$ |
| Unknown Questions | – | $N_4$ | $N_5$ |

correctly answering questions within its knowledge boundary, and truthful, by abstaining from those outside it. To formally measure progress towards this goal, we establish an evaluation framework that assesses the performance of a refined LLM relative to the initial LLM before fine-tuning. As illustrated in the abstention confusion matrix in Table 1, we partition the evaluation benchmark into two distinct categories based on the performance of the initial LLM:

- **Known Questions**: The set of questions that the initial LLM answers correctly.
- **Unknown Questions**: The set of questions that the initial LLM answers incorrectly.

The refined LLM is then evaluated on its ability to preserve accuracy on known questions while learning to abstain from unknown questions.

## 2.2 RELIABILITY EVALUATION

### 2.2.1 HELPFULNESS AND TRUTHFULNESS

Helpfulness evaluates a model's capacity to preserve accuracy on questions within its known scope, preventing unnecessary abstention and forgetting. Truthfulness measures its ability to abstain from questions outside that scope, preventing fabricated responses. Following Kim et al. (2024); Feng et al. (2024), we adopt two fine-grained F1 metrics $\text{F1}_{ans}$ for helpfulness evaluation and $\text{F1}_{abs}$ for truthfulness evaluation:

$$\text{F1}_{ans} = \frac{2 \cdot \text{Precision}_{ans} \cdot \text{Recall}_{ans}}{\text{Precision}_{ans} + \text{Recall}_{ans}} = \frac{2N_1}{2N_1 + 2N_2 + N_3 + N_4}, \quad (1)$$

$$\text{Precision}_{ans} = \frac{N_1}{N_1 + N_2 + N_4}, \quad \text{Recall}_{ans} = \frac{N_1}{N_1 + N_2 + N_3}, \quad (2)$$

$$\text{F1}_{abs} = \frac{2 \cdot \text{Precision}_{abs} \cdot \text{Recall}_{abs}}{\text{Precision}_{abs} + \text{Recall}_{abs}} = \frac{2N_5}{N_3 + N_4 + 2N_5}, \quad (3)$$

$$\text{Precision}_{abs} = \frac{N_5}{N_3 + N_5}, \quad \text{Recall}_{abs} = \frac{N_5}{N_4 + N_5}. \quad (4)$$

### 2.2.2 RELIABILITY METRIC

Since increasing truthfulness often leads to a decrease in helpfulness, using either metric alone introduces bias. Therefore, we propose $\text{F1}_{rel}$, defined as the harmonic mean of $\text{F1}_{ans}$ and $\text{F1}_{abs}$, which provides a holistic measure of reliability:

$$\text{F1}_{rel} = \frac{2 \cdot \text{F1}_{ans} \cdot \text{F1}_{abs}}{\text{F1}_{ans} + \text{F1}_{abs}} = \frac{4N_1 N_5}{4N_1 N_5 + 2N_2 N_5 + N_1 N_3 + N_1 N_4 + N_3 N_5 + N_4 N_5}. \quad (5)$$

The $\text{F1}_{rel}$ metric holistically considers all categories in the abstention confusion matrix in Table 1 and exhibits the following desiderata: it monotonically increases with the correctly answered known questions ($N_1$) and correct abstention from unknown questions ($N_5$); it decreases monotonically with both types of error: incorrect answer ($N_2$, $N_3$) and over-abstention ($N_4$), and therefore is intuitively reasonable. In the ideal case ($N_2 = N_3 = N_4 = 0$, $N_1 + N_5 = N$), $\text{F1}_{rel} = 1$; in the worst case ($N_1 = N_5 = 0$, $N_2 + N_3 + N_4 = N$), $\text{F1}_{rel} = 0$, where $N$ is the number of all questions. This metric effectively balances helpfulness and truthfulness, mitigating the risks of overly conservative or aggressive model behavior. A recent metric, reliability score (RS) (Xu et al., 2024a), has been proposed to achieve similar balancing effect, but we show that it is flawed in encouraging hallucination (see Appendix C.1).

## 3 METHOD

Inspired by the powerful generalization capabilities of reinforcement learning (Chu et al., 2025) and its success in reasoning (Guo et al., 2025; Zuo et al., 2025), we incorporate abstention decisions into the rollout process and train LLMs to abstain appropriately via fine-grained confidence reward (FISCORE). In this section, we first show the details of training FISCORE and formalize the reinforcement learning objective. Then, we demonstrate the prompt template that guides the LLMs to generate rollout in the defined format. Finally, we introduce the reward modeling to guide the optimization of reinforcement learning to improve reliability.

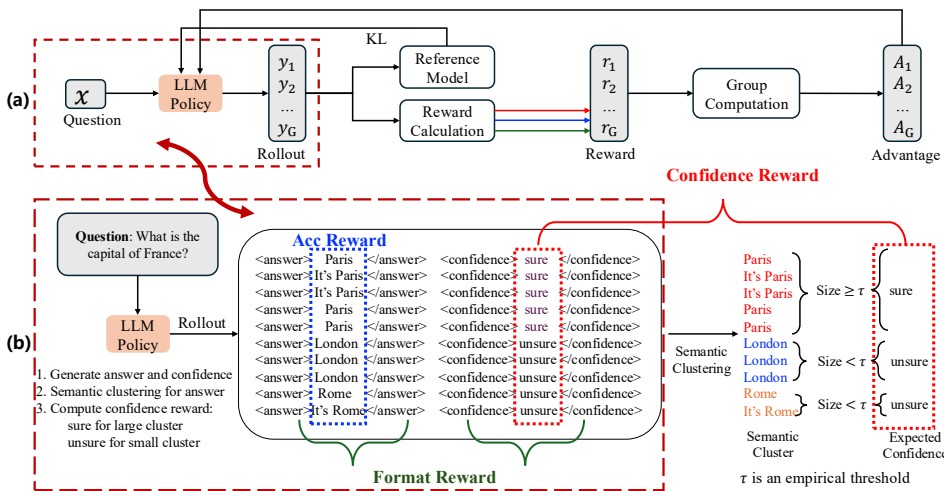

Figure 2: The training overview of our method. (a) The GRPO pipeline. (b) A detailed example of how FISCORE works.

### 3.1 GROUP RELATIVE POLICY OPTIMIZATION

We reward the model based on the sample-specific semantic confidence among a group of its own sampled responses, which requires a reinforcement learning algorithm capable of handling multiple samples per prompt. Group Relative Policy Optimization (GRPO) (Shao et al., 2024) naturally fits this setting: it samples a group of $G$ outputs (or *rollouts*) for each question $x$, which directly supports our semantic clustering–based confidence reward computation. Formally, for each question $x$, GRPO samples a group of $G$ outputs $Y = \{y_1, y_2, \ldots, y_G\}$ from the old policy $\pi_{\theta_{old}}$ and optimizes the new policy $\pi_\theta$ by maximizing the following objective:

$$\mathcal{J}_{GRPO}(\theta) = \mathbb{E}_{[x \sim P(Q), \{y_i\}_{i=1}^G \sim \pi_{\theta_{old}}(Y|x)]}$$

$$\frac{1}{G} \sum_{i=1}^{G} \left( \min \left( \frac{\pi_\theta(y_i|x)}{\pi_{\theta_{old}}(y_i|x)} A_i, \text{clip} \left( \frac{\pi_\theta(y_i|x)}{\pi_{\theta_{old}}(y_i|x)}, 1 - \epsilon, 1 + \epsilon \right) A_i \right) - \beta \mathbb{D}_{KL}(\pi_\theta || \pi_{ref}) \right),$$

$$(6)$$

where $A_i = \left( r_i - \text{mean}(\{r_i\}_{i=1}^G) \right) / \text{std}(\{r_i\}_{i=1}^G)$ is the normalized advantage of the $i$th rollout in the current group, $\epsilon$ is the clipping ratio, and $\beta$ is the KL loss coefficient. Moreover, the objective also incorporates a KL divergence penalty, specifically to prevent excessive divergence of the policy from the original reference policy model. The GRPO pipeline is shown in Figure 2 (a).

### 3.2 TRAINING TEMPLATE

As we describe the confidence reward in Figure 2 (b), it is essential for the policy LLM to generate responses that conform to the specified format. To facilitate this, we design a prompt template to guide the policy model in producing an answer along with an associated confidence level. As shown in Table 2, the model is instructed to format its responses into two distinct parts. First, the model

Table 2: Training template. The question is appended at the end during training and inference.

---

You are a helpful and truthful AI Assistant. Given a question, you should provide responses that include answer and confidence. You first answer the question as briefly as possible enclosed by `<answer>` and `</answer>`, and then provide your confidence in sure or unsure about the answer, enclosed by `<confidence>` and `</confidence>`. Respond in the following format:
`<answer>` ... `</answer>`
`<confidence>` sure or unsure `</confidence>`
Question:

---

provides the answer wrapped in the `<answer>` `</answer>` tag. Second, the model expresses a binary confidence level (sure or unsure) wrapped in the `<confidence>` `</confidence>` tag. This `<answer>` + `<confidence>` structure enforces the model to reflect on its confidence after providing an answer. In this work, we interpret an unsure confidence as an act of abstention.

## 3.3 REWARDS FUNCTION DESIGN

Figure 2 (b) illustrates the proposed FISCORE. We first partition the responses using a proxy model that captures semantic equivalence, then introduce the confidence reward and auxiliary rewards.

### 3.3.1 FINE-GRAINED SEMANTIC CONFIDENCE REWARD

Some studies employ reinforcement learning with confidence rewards to enhance the reasoning performance of LLMs (Zuo et al., 2025; Shafayat et al., 2025). We extend this idea to the abstention domain, which guides LLMs to answer when confident and abstain otherwise.

**Semantic Clustering.** To compute the semantic confidence reward for individual rollouts during GRPO training, we initially organize responses that express equivalent meanings through semantic clustering (Kuhn et al., 2023). These semantic clusters represent equivalence classes derived from a semantic equivalence relation that satisfies reflexivity, symmetry, and transitivity properties while capturing the semantic similarity between textual inputs. The practical implementation of equivalence via bidirectional entailment predictions generated by a Natural Language Inference (NLI) model, specifically DeBERTa (He et al., 2021), which determines the relationships between text pairs as either entailment, neutral, or contradiction. Texts are considered semantically equivalent when they mutually entail one another in both directions. We follow Kuhn et al. (2023) employing a greedy algorithm that assigns each response to an existing semantic cluster if it demonstrates semantic equivalence with any member of that cluster.

**Semantic Confidence**. The formulation of FISCORE is based on the intuition that generative semantic consensus reflects model confidence. We assume that a larger semantic cluster implies strong consensus among the generated responses, reflecting higher confidence. Conversely, smaller or singleton clusters suggest divergent or less-supported meanings, thus implying lower confidence. Following this principle, we quantify the confidence of each rollout using a measure derived from the cardinality of the semantic cluster it belongs to. This per-sample approach distinguishes our method from previous work that relies on a single, global metric like semantic entropy calculated over all responses. For a given rollout $y_i$ and its corresponding semantic cluster $C_i$, the confidence reward function is defined as:

$$R_c = \begin{cases} 1, & \text{if } (|C_i| \geq \tau \text{ and } U = \text{"sure"}) \text{ or } (|C_i| < \tau \text{ and } U = \text{"unsure"}) \\ 0, & \text{otherwise} \end{cases} \quad (7)$$

Here, $\tau$ is an empirical threshold of abstention, which we set to $\lceil \frac{G}{2} \rceil$ in all experiments, where $G$ is the total number of rollouts. $U$ is the model's expressed confidence, either "sure" or "unsure", extracted from the `<confidence>` `</confidence>` tag within the rollout. By incentivizing this alignment, the reward function effectively teaches the model a reliable abstention policy: to answer when semantic confidence is high and to abstain otherwise.

### 3.3.2 Auxiliary Rewards

**Accuracy Reward.** To prevent the model from engaging in reward hacking by learning to maximize the confidence reward on a uniform but incorrect answer, we introduce an auxiliary accuracy reward. This reward is granted only when the answer is semantically equivalent to the ground truth solution.

$$R_a = \begin{cases} 1, & \text{if the answer and the solution are semantically equivalent} \\ 0, & \text{otherwise} \end{cases} \tag{8}$$

where the answer is extracted between `<answer>` `</answer>` tags.

**Format Reward.** We encourage the model to enclose the answer within `<answer>` `</answer>` tags and to present the confidence enclosed within `<confidence>` `</confidence>` tags. The format reward $R_f$ is composed of two components. For each correctly used tag (`<answer>`, `</answer>`, `<confidence>`, `</confidence>`) that appears in the rollout, we assign a reward of 0.125. To allow the model to determine its confidence based on its answer, it is necessary to place the answer before the confidence expression. We assign an additional reward of 0.5 if the tags appear in the correct order.

The total reward $R_{total}$ is defined as a weighted sum of three distinct components:

$$R_{total} = w_c R_c + w_a R_a + w_f R_f, \tag{9}$$

where $w_c$, $w_a$, $w_f$ are the respective weights for each reward component. Notably, only when the format is correct ($R_f = 1$) will the other two rewards be added.

## 4 Experiments

### 4.1 Experiment Setting

**Datasets.** We use four datasets of question-answering (QA) task for evaluation, including **Pararel** (Elazar et al., 2021) which contains cloze-style questions of relation prediction, **TriviaQA** (Joshi et al., 2017) which contains general knowledge QA pairs, **Natural Questions (NQ)** (Kwiatkowski et al., 2019) which contains questions from aggregated queries to Google Search, and **SciQ** (Welbl et al., 2017) which contains science exam questions. We use the **Pararel** dataset for training.

**Baselines.** We compare FISCORE with three types of baselines.

1. **Prompting method**: In-Context Learning (ICL), In-Context Learning with refusal examples (ICL-IDK), and In-Context Learning with confidence expression (ICL-Unsure). We take the performance of ICL as the performance of the initial model.

2. **Supervised fine-tuning methods**: R-Tuning (Zhang et al., 2024), R-Tuning-U (Zhang et al., 2024), and SE-Tuning (Tjandra et al., 2024).

3. **RL-based method**: GRPO with semantic entropy (GRPO-SE), which adopts the training data as SE-Tuning and is built on GRPO.

**Evaluation.** We use four metrics: $F1_{ans}$, $F1_{abs}$, $F1_{rel}$, and accuracy (Acc) for evaluation, which are detailed in Section 2.2. We evaluate the correctness of the model's response by prompting with 6-shot examples. Following Xue et al. (2025), we adopt bidirectional string matching to evaluate answer correctness. Here, we treat a response as an abstention if the confidence part is unsure. More details about the evaluation are listed in Appendix D.

**Implementation Details.** We choose Llama3-8B-Instruct (Grattafiori et al., 2024) and Qwen2.5-7B-Instruct (Yang et al., 2024) in our experiments. The reinforcement learning framework is built on Open-R1 (Face, 2025). We set the reward weights $w_c = 1$, $w_a = 4$ and $w_f = 2$. We use Deberta-v2-xlarge-mnli (He et al., 2021) as the NLI model for semantic clustering. The sampling number is set to 10, and the threshold $\tau$ is set to 5. All experiments are implemented on Nvidia L40-48GB GPUs. The training epoch is set to 1. We set the sampling temperature to 1.0 during the GRPO training. During inference, we utilize the vLLM framework to accelerate the process and employ a greedy search strategy to generate responses. Hyperparameters and additional configurations are detailed in Appendix B.

Table 3: Performance on in-domain and out-of-domain knowledge-intensive QA datasets. All results are multiplied by 100. Notably, ICL is the initial model, so we do not calculate its F1 scores.

| Method | Pararel (ID) | | | | TriviaQA (OOD) | | | | NQ (OOD) | | | | SciQ (OOD) | | | |
|---|---|---|---|---|---|---|---|---|---|---|---|---|---|---|---|---|
| | $F1_{ans}$ | $F1_{abs}$ | $F1_{rel}$ | Acc | $F1_{ans}$ | $F1_{abs}$ | $F1_{rel}$ | Acc | $F1_{ans}$ | $F1_{abs}$ | $F1_{rel}$ | Acc | $F1_{ans}$ | $F1_{abs}$ | $F1_{rel}$ | Acc |
| **Llama3-8B-Instruct** | | | | | | | | | | | | | | | | |
| ICL | - | - | - | **48.8** | - | - | - | **68.4** | - | - | - | **37.1** | - | - | - | **65.3** |
| ICL-IDK | 62.1 | 64.3 | 63.2 | 31.7 | 69.8 | 32.6 | 44.5 | 55.3 | 43.5 | 67.9 | 53.0 | 18.1 | 71.8 | 43.9 | 54.5 | 56.4 |
| ICL-Unsure | 70.4 | 68.2 | 69.3 | 41.9 | 67.7 | 46.4 | 55.0 | 51.7 | 45.9 | 18.0 | 25.9 | 30.4 | 64.7 | 17.9 | 28.0 | 53.1 |
| R-Tuning | 78.3 | **79.6** | **79.0** | 40.5 | 47.7 | 51.5 | 49.5 | 26.3 | 31.2 | 72.0 | 43.5 | 10.8 | 32.1 | 51.4 | 39.5 | 14.6 |
| R-Tuning-U | 73.8 | 76.6 | 75.2 | 40.5 | 42.6 | 53.2 | 47.3 | 20.4 | 31.8 | **74.0** | 44.5 | 10.3 | 49.9 | 49.7 | 49.8 | 28.0 |
| SE-Tuning | 78.2 | 75.0 | 76.6 | 45.9 | 44.9 | 50.7 | 47.7 | 24.0 | 27.7 | 72.2 | 40.0 | 9.4 | 22.4 | 50.4 | 31.0 | 9.2 |
| GRPO-SE | 70.9 | 77.3 | 73.9 | 35.1 | 70.5 | 60.2 | 65.0 | 47.3 | 37.2 | 73.9 | 49.5 | 13.0 | 64.4 | 51.2 | 57.0 | 46.4 |
| FISCORE | **83.3** | 67.1 | 74.3 | **59.4** | **82.6** | 62.0 | 70.8 | 61.7 | **53.1** | 72.5 | **61.3** | 21.7 | **74.9** | 53.3 | 62.3 | 52.4 |
| **Qwen2.5-7B-Instruct** | | | | | | | | | | | | | | | | |
| ICL | - | - | - | 44.1 | - | - | - | **51.7** | - | - | - | **27.6** | - | - | - | **68.8** |
| ICL-IDK | 60.1 | 68.6 | 64.1 | 27.4 | 66.6 | 49.3 | 56.6 | 44.9 | 38.1 | 50.1 | 43.3 | 19.4 | 71.6 | 14.8 | 24.5 | 60.5 |
| ICL-Unsure | 77.7 | 82.9 | 80.2 | 35.8 | 71.2 | 51.7 | 59.9 | 49.8 | 40.2 | 39.7 | 39.9 | 21.9 | 71.4 | 37.6 | 49.2 | 57.3 |
| R-Tuning | 79.3 | **86.4** | **82.7** | 31.8 | 44.0 | 71.0 | 54.4 | 15.1 | 25.1 | 84.5 | 38.7 | 4.5 | 43.8 | 53.9 | 48.3 | 19.9 |
| R-Tuning-U | 62.1 | 80.5 | 70.1 | 21.3 | 40.1 | 69.8 | 51.0 | 13.6 | 32.3 | **84.8** | 46.7 | 6.3 | 43.9 | 52.5 | 47.8 | 20.4 |
| SE-Tuning | 71.4 | 82.3 | 76.5 | 28.0 | 33.8 | 69.3 | 45.4 | 10.7 | 32.8 | 84.2 | 47.2 | 6.7 | 55.4 | 52.7 | 54.0 | 34.5 |
| GRPO-SE | 75.2 | 83.3 | 79.0 | 31.7 | 71.6 | 72.4 | 72.0 | 41.6 | 42.8 | 78.0 | 55.3 | 14.2 | 76.7 | 50.7 | 61.0 | 57.3 |
| FISCORE | **80.8** | 72.9 | 76.6 | **51.9** | **83.3** | 79.1 | **81.1** | 51.3 | **57.3** | 80.1 | **66.8** | 20.4 | **79.2** | 54.5 | **64.6** | 58.0 |

## 4.2 Main Results

We present the main experimental results of FISCORE and all baseline methods in Table 3, and we show the methods based on different LLMs. The results reveal several key findings:

**Generalizable Reliability.** We focus on metric $F1_{rel}$ that assesses a model's overall reliability by balancing helpfulness ($F1_{ans}$) and truthfulness ($F1_{abs}$). A key finding is that the high reliability demonstrated by some baselines on in-domain (ID) data is brittle and does not generalize. For example, R-Tuning using the Llama model achieves a high $F1_{rel}$ score of 79.0 on the Pararel (ID) dataset by directly learning to abstain from unknown questions. However, this specialized performance collapses on out-of-distribution (OOD) tasks, with its score dropping to just 49.5 of $F1_{rel}$ in TriviaQA. In contrast, the proposed FISCORE demonstrates a more robust, reliable performance that excels at generalization. It achieves this by attaining a superior balance between answering known questions ($F1_{ans}$ of 83.3) and correctly abstaining ($F1_{abs}$ of 67.1). This balanced approach leads to a high $F1_{rel}$ score of 74.3 on the ID dataset, which shows a much smaller drop to 70.8 on the OOD TriviaQA dataset. This consistent outperformance on all three OOD datasets proves that the knowledge boundary learned by our approach is more generalizable. Compared to SE-Tuning, GRPO-SE shows stronger generalization: SE-Tuning performs well on ID data but degrades significantly on OOD sets, while GRPO-SE maintains robust performance across both, highlighting the advantage of reinforcement learning over supervised fine-tuning.

**Effectiveness of the Fine-grained Semantic Confidence Reward.** We compare our approach directly with GRPO-SE, a baseline employing the same GRPO algorithm but utilizing a coarse semantic uncertainty reward. The results demonstrate a clear advantage: our method FISCORE significantly outperforms GRPO-SE across all OOD datasets and different LLMs in terms of $F1_{rel}$. Given that the primary distinction between the two methods is the reward function, this comparison provides empirical evidence that our fine-grained confidence reward is substantially more effective than a coarse, global entropy-based signal.

**Comparison between Different LLMs.** We fine-tuned both the Llama3-8B-Instruct and Qwen2.5-7B-Instruct models on the Pararel dataset using our FISCORE method alongside established baseline abstention techniques. From the results on TriviaQA and NQ we can see that although Llama

achieved higher raw accuracy than Qwen both before and after fine-tuning, its overall reliability (measured by $F1_{rel}$) remained lower. This disparity indicates that a model's extensive knowledge capacity does not ensure precise self-awareness of its knowledge boundaries. It also highlights that accuracy alone provides an inadequate proxy for gauging improvements in abstention alignment.

### 4.3 Ablation Study

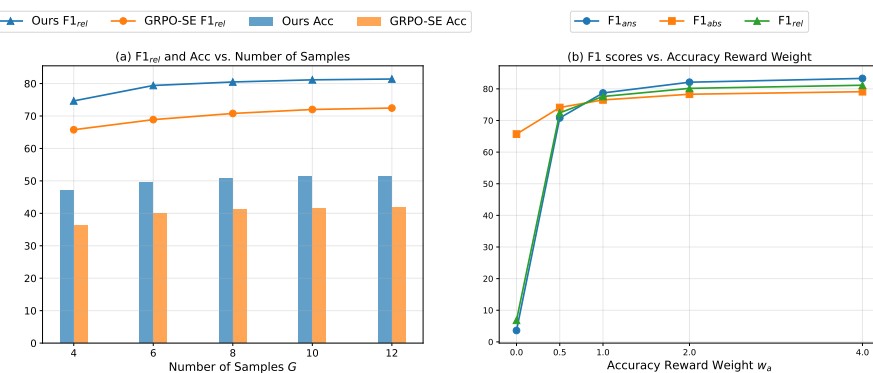

Figure 3: (a) Experiments of $F1_{rel}$ and accuracy of various sampling number $G$ on TriviaQA on Qwen2.5-7B-Instruct. (b) Experiments of $F1_{ans}$, $F1_{abs}$ and $F1_{rel}$ of various accuracy reward weight $w_a$ on TriviaQA on Qwen2.5-7B-Instruct.

**Impact of Sampling Number.** We investigate the impact of the sampling number $G$ on performance in Figure 3 (a). The results demonstrate that for our method FISCORE, both reliability ($F1_{rel}$) and accuracy improve as $G$ increases, consistently outperforming the GRPO-SE baseline. Both metrics reach their optimal or near-optimal levels at $G = 10$. Since performance gains become negligible beyond this point, we select $G = 10$ as the default setting for all experiments to maximize performance while maintaining computational efficiency.

**Impact of Accuracy Reward Weight.** We evaluate the impact of the accuracy reward weight $w_a$ while fixing the confidence reward weight $w_c$ at 1. When $w_a$ is set to 0, the model achieves a high $F1_{abs}$ but a near-zero $F1_{ans}$, indicating that the model abstains from almost all questions. This occurs because the model receives rewards for generating low-confidence incorrect answers, a phenomenon known as reward hacking. Shafayat et al. (2025) demonstrate similar results: relying on self-confidence as the only reward leads to model collapse in the later stages of training. Therefore, it is necessary to combine accuracy reward and confidence reward for training. Introducing the accuracy reward improves $F1_{rel}$. As the accuracy reward weight increases, both $F1_{ans}$ and $F1_{rel}$ rise steadily. Notably, weights greater than 1 produce higher $F1_{rel}$ scores, likely because obtaining accuracy rewards is more difficult than obtaining confidence rewards, thus requiring greater emphasis.

Table 4: Ablation on threshold.

|   | $\mathbf{F1}_{ans}$ | $\mathbf{F1}_{abs}$ | $\mathbf{F1}_{rel}$ | $Acc$ |
|---|------|------|------|------|
| 3 | 79.3 | 65.6 | 71.8 | 54.7 |
| 4 | 82.4 | 75.4 | 78.8 | 53.0 |
| 5 | 83.3 | 79.1 | 81.1 | 51.3 |
| 6 | 82.1 | 78.3 | 80.1 | 49.9 |
| 7 | 74.7 | 81.2 | 77.8 | 36.5 |

Table 5: Ablation on reward functions.

|   | $\mathbf{F1}_{ans}$ | $\mathbf{F1}_{abs}$ | $\mathbf{F1}_{rel}$ | $Acc$ |
|---|------|------|------|------|
| FISCORE | 83.3 | 79.1 | 81.1 | 51.3 |
| w/o Format reward | 71.5 | 43.6 | 54.2 | 51.5 |
| w/o Confidence reward | 69.5 | 2.1 | 4.1 | 56.1 |
| w/ Continuous reward | 83.0 | 82.9 | 82.9 | 49.1 |

**Impact of Abstention Threshold.** We conduct ablation experiments on the TriviaQA dataset using the Qwen2.5-7B-Instruct model to investigate the impact of the abstention threshold $\tau$. As shown in the Table 4, $F1_{abs}$ increases with rising $\tau$, while $F1_{ans}$ and Acc exhibit a downward trend. At $\tau = 5$, truthfulness and helpfulness achieve a balanced level, yielding the maximum $F1_{rel}$. Therefore, different thresholds can be employed for training across scenarios with varying tolerance levels for hallucinations.

**Ablation on Reward Functions.** Table 5 presents the results of reward function ablation experiments conducted on the TriviaQA dataset using the Qwen2.5-7B-Instruct model. We first validate the necessity of the format reward. We remove the format reward and observe a decrease in $F1_{ans}$, a significant drop in $F1_{abs}$, while accuracy remains almost unaffected. The reason lies in semantic clustering: we extract answers within the answer tags using rule-based methods. The lack of format reward leads to inaccurate tags, which degrade semantic clustering performance and ultimately increase confidence bias. To validate the effectiveness of the proposed fine-grained semantic confidence reward, we removed this reward and observed that while accuracy remained stable, $F1_{abs}$ dropped to near zero. This demonstrates that the model possesses a certain degree of confidence perception capability, and our confidence reward can significantly enhance this ability. Finally, we train Qwen2.5-7B-Instruct with continuous confidence rewards, where the model samples 10 candidate answers and predicts confidence scores scaled from 10% to 100%. We find that continuous rewards achieve better reliability than binary discrete rewards. Continuous rewards do not require retraining when adapting the abstention threshold, resulting in lower applied costs.

## 4.4 DETAILED ANALYSIS

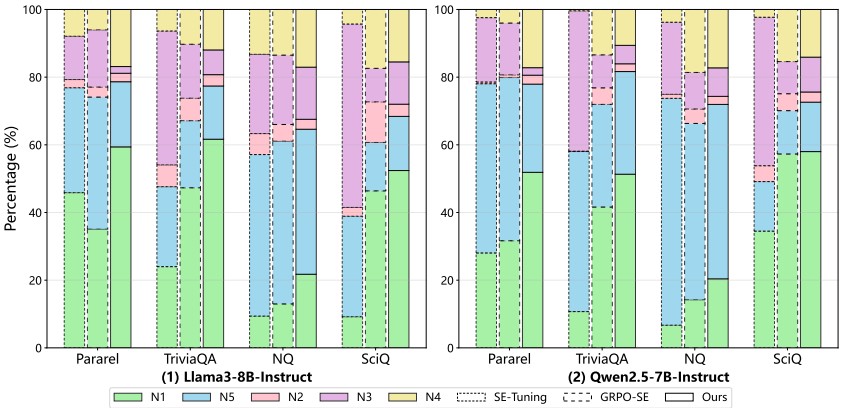

Figure 4: Percentage of prediction types among different methods. We choose SE-Tuning, GRPO-SE, and FISCORE.

Figure 4 compares the performance of three methods: SE-Tuning, GRPO-SE, and FISCORE. The primary goal is to maximize correctly answered known questions ($N_1$) and abstained unknown questions ($N_5$), while minimizing errors ($N_2$, $N_4$) and unnecessary abstentions ($N_3$). Note that Pararel is the ID dataset, whereas all the others are OOD. On OOD datasets, SE-Tuning tends to over-abstain, reflected by a larger $N_3$, indicating a conservative strategy that suppresses hallucinations at the cost of unnecessary refusals on answerable cases. By contrast, GRPO-SE and FISCORE show higher hallucination rates than SE-Tuning, which reflects the seesaw effect between helpfulness and truthfulness: reducing over-abstention encourages the model to answer more often and thus increases the chance of hallucinations relative to an overly conservative baseline. Importantly, in most cases FISCORE attains a higher $N_1+N_5$ than the baselines, indicating a better overall balance between answering known questions and abstaining on unknown ones, and hence higher reliability.

## 5 RELATED WORK

**Uncertainty Estimation and Confidence Elicitation.** Uncertainty estimation aims to quantify this risk before LLMs respond to a request, which is an effective tool for detecting and mitigating hallucinations of LLMs (Xia et al., 2025). Likelihood-based methods estimate uncertainty by computing probabilities for salient tokens, capturing the model's confidence in specific outputs (Duan et al., 2024; Lin et al., 2024). Sampling-based methods generate multiple candidate responses, cluster them, and quantify uncertainty based on the degree of inter-response consistency (Kuhn et al., 2023; Qiu & Miikkulainen, 2024; Nikitin et al., 2024; Aichberger et al., 2025). Probing-based methods leverage internal model representations to train classifiers that predict uncertainty or detect hallucinations, often without requiring additional generation (Han et al., 2024; Marks & Tegmark, 2024;

Gottesman & Geva, 2024; Azaria & Mitchell, 2023). Confidence elicitation aims to elicit the confidence of LLMs in their answers (Geng et al., 2024). Prompting methods directly prompt LLMs to express confidence (Lin et al., 2022; Kadavath et al., 2022). Recent studies (Xu et al., 2024b; Damani et al., 2025) employ chain-of-thought (CoT) training to teach LLMs to reason about their confidence. Zheng et al. (2025) adopt binary confidence elicitation ("sure" vs. "unsure") as a proxy for abstention, enabling models to discard low-confidence answers post hoc. In this work, we introduce a fine-grained semantic confidence reward to cultivate a more nuanced self-awareness of the model's confidence.

**Abstention Fine-Tuning.** Many studies explore fine-tuning LLMs to abstain from answering questions beyond their knowledge boundaries (Wen et al., 2025). Brahman et al. (2024); Cheng et al. (2024); Xu et al. (2024a) propose learning from preferences via direct preference optimization (Rafailov et al., 2023) to train LLMs to abstain from answering unknown questions. Li et al. (2025b) introduce adaptive contrastive preference learning to calibrate abstention behavior. Zhang et al. (2024); Tjandra et al. (2024); Xue et al. (2025) leverage uncertainty to construct abstention-aware datasets, replacing the ground-truth labels of uncertain questions with "I don't know". Yang et al. (2025); Deng et al. (2024) train LLMs to reason about their own uncertainty and provide post-hoc explanations for why a given question is unanswerable. Huang et al. (2025); Cohen et al. (2024) introduce a special "rejection" token into the model's vocabulary and design an objective function that reallocates probability mass toward the "rejection" token when the model encounters uncertain predictions. In this work, we go beyond aggregated uncertainty by utilizing fine-grained, per-sample confidence to sharpen the model's awareness of its knowledge limits.

## 6 CONCLUSION

In this paper, we focus on enhancing the reliability of LLMs, which we define as the ability to accurately answer questions within their knowledge boundaries and abstain from those beyond them. First, we propose a new comprehensive reliability metric by calculating the harmonic mean of the F1 scores for answering and abstaining. This metric integrates model helpfulness and truthfulness into a single, robust score that is monotonically sensitive to all error categories. We then introduce FISCORE, a reinforcement learning method with a fine-grained semantic confidence reward. This reward mechanism, derived from the semantic confidence among multiple generated responses, guides the model to retain the answer only when generative consensus is high and to discard it when it is low, thereby enabling the model to make abstention decisions more aligned with its knowledge boundaries. Our experiments on both in-domain and out-of-distribution datasets show that FISCORE achieves a better balance between helpfulness and truthfulness than baselines, thereby improving overall model reliability.

## REPRODUCIBILITY STATEMENT

To ensure reproducibility of our work, we provide detailed instructions and the necessary code. The source code for training and evaluation is available in the supplementary material. We also provide the baseline code.

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

## A  THE USE OF LARGE LANGUAGE MODELS

We use LLMs to refine the usage of terminology and sentences. We also use LLMs to adjust paragraph structure in the main body of this paper. We also use LLMs to search for related works.

## B  EXPERIMENT DETAILS

### B.1  DATASET

We test on the following publicly available datasets.

- **Pararel** (Elazar et al., 2021) is a dataset of factual knowledge and relations that are originally for evaluating masked language models. We follow Zhang et al. (2024), changing it into a question-answering format for generation evaluation. The validation set contains 5,584 QA pairs, which we use for evaluation.
- **TriviaQA** (Joshi et al., 2017) is a large-scale reading comprehension dataset containing over 95,000 trivia-style question–answer pairs. The validation set contains 11,313 QA pairs, which we use for evaluation.
- **Natural Questions (NQ)** (Kwiatkowski et al., 2019) are aggregated queries issued to the Google search engine. All questions of NQ can be answered using the contents of the English Wikipedia. The validation set contains 3,610 QA pairs, which we use for evaluation.
- **SciQ** (Welbl et al., 2017) is a crowdsourced dataset of 13,679 science questions. It covers topics from biology, chemistry, physics, and earth science from real educational materials. The validation set contains 1,000 QA pairs, which we use for evaluation.

### B.2  DETAILS OF BASELINES

The core idea of existing abstention methods is to fine-tune LLMs to answer the questions within their knowledge scope and abstain from those outside it. It consists of two steps.

First, given a training dataset $D$, these methods partition it into two subsets, the answerable set $D_0$ and the unanswerable set $D_1$. For questions in $D_0$, the model is assumed to be confident about it, and the ground-truth answer is set to be the model's standard response. For questions in $D_1$, the ground-truth text, it is changed to an abstention expression: "I don't know". R-Tuning (Zhang et al., 2024) prompts LLMs to answer questions from the training set, then classifies correctly answered questions into $D_0$ and incorrectly answered ones into $D_1$. R-Tuning-U (Zhang et al., 2024) and SE-Tuning (Tjandra et al., 2024) use uncertainty to partition the training set. R-Tuning adopts predictive entropy, which clusters lexical equivalent answers together. SE-Tuning adopts semantic entropy, which clusters semantically equivalent answers together. Let $C_1, ..., C_m$ be the classes that were clustered from sampled responses; the entropy is given by:

$$Entropy(\mathbf{x}) \approx -\sum_{i=1}^{m} p_\theta(C_i \mid \mathbf{x}) \log p_\theta(C_i \mid \mathbf{x}) \approx -\sum_{i=1}^{m} \left( \frac{|C_i|}{M} \right) \log \left( \frac{|C_i|}{M} \right), \quad (10)$$

where $M$ is the total number of samples. Then, $D_0$ and $D_1$ is obtained by a threshold $\tau$. Then, LLMs are fine-tuned on $D_0$ and $D_1$, learning to minimize the following objective:

$$\mathcal{L}_{CE}(p_\theta) = -\sum_{x \in D_0, D_1} \sum_{t=1}^{|\mathbf{y}^{(x)}|} \log p_\theta(y_t^{(x)} \mid \text{prompt}, \mathbf{x}, \mathbf{y}_{<t}^{(x)}), \quad (11)$$

For consistency, GRPO-SE is trained on the same processed dataset as SE-Tuning. Its objective function is Equation 6, enabling a controlled comparison of reward granularity and abstention behavior.

### B.3  TRAINING DETAILS

We use 15,000 samples from the Pararel dataset for training. Before training, we remove questions with semantic entropy below 1. These low-entropy samples lack sufficient semantic diversity, which

can bias the binary confidence reward and lead the model to abstain too infrequently. Moreover, their simplicity diminishes the relative advantage signal in GRPO training, reducing the effectiveness of policy optimization. In the semantic clustering, we concatenate the question and the answer as the input of the NLI model. we run each experiment with three different random seeds and report the mean performance.

### B.4 HYPERPARAMETERS

For FISCORE on Llama3-8B-Instruct and Qwen2.5-7B-Instruct, we use the same hyperparameter setups, which are shown in Table 6.

Table 6: Hyperparameter setups for GRPO trainer.

| Parameter | Value |
|---|---|
| *General Settings* | |
| bf16 | true |
| use_vllm | true |
| vllm_device | auto |
| vllm_enforce_eager | true |
| vllm_gpu_memory_utilization | 0.7 |
| vllm_max_model_len | 4608 |
| do_eval | false |
| *Training Configuration* | |
| gradient_accumulation_steps | 4 |
| gradient_checkpointing | true |
| gradient_checkpointing_kwargs | use_reentrant: false |
| learning_rate | 3.0e-06 |
| lr_scheduler_type | cosine_with_min_lr |
| lr_scheduler_kwargs | min_lr_rate: 0.1 |
| warmup_ratio | 0.05 |
| num_train_epochs | 1 |
| per_device_train_batch_size | 10 |
| per_device_eval_batch_size | 10 |
| threshold of abstention | 5 |
| *Generation Settings* | |
| max_prompt_length | 256 |
| max_completion_length | 64 |
| num_generations | 10 |
| temperature | 1.0 |
| seed | 42, 32, 22 |
| *Logging and Saving* | |
| log_completions | true |
| log_level | info |
| logging_first_step | true |
| logging_steps | 1 |
| logging_strategy | steps |
| save_strategy | steps |
| save_steps | 50 |
| report_to | wandb |
| *Reward Configuration* | |
| reward_funcs | confidence, format, accuracy (semantic) |
| reward_weights | 1.0 2.0, 4.0 |

## C  RELIABILITY METRIC

### C.1  ANALYSIS FOR RELIABILITY SCORE

Xu et al. (2024a) propose the Reliability Score (RS), which encourages LLMs to offer maximal assistance while minimizing hallucinated errors. RS is defined as a weighted sum of accuracy and truthful rate, which is used by current studies (Xu et al., 2024a; Kim et al., 2025; Yang et al., 2025): $RS(\alpha) = \alpha \times \text{Truth.} + (1 - \alpha) \times \text{Acc.}$, the weight $\alpha$ is answering rate.

The components are:

$$N = N_1 + N_2 + N_3 + N_4 + N_5$$

$$\text{Acc.} = \frac{N_1}{N}$$

$$\text{Truth.} = \frac{N_1 + N_3 + N_5}{N}$$

$$\alpha = 1 - \frac{N_3 + N_5}{N} = \frac{N_1 + N_2 + N_4}{N}$$

Substituting these into the RS formula:

$$RS = \left( \frac{N_1 + N_2 + N_4}{N} \right) \left( \frac{N_1 + N_3 + N_5}{N} \right) + \left( 1 - \frac{N_1 + N_2 + N_4}{N} \right) \left( \frac{N_1}{N} \right)$$

$$= \left( \frac{N_1 + N_2 + N_4}{N} \right) \left( \frac{N_1 + N_3 + N_5}{N} \right) + \left( \frac{N_3 + N_5}{N} \right) \left( \frac{N_1}{N} \right)$$

$$= \frac{(N_1 + N_2 + N_4)(N_1 + N_3 + N_5) + N_1(N_3 + N_5)}{N^2}$$

The partial derivative of RS with respect to $N_4$ is:

$$\frac{\partial \text{RS}}{\partial N_4} = \frac{N_3 + N_5 - (N_1 + N_2 + N_4)}{N^2} \tag{12}$$

The sign of this derivative is variable and contingent on the specific distribution of counts among the categories. It is not guaranteed to be negative. For instance, consider a perfect model evaluated on a dataset composed solely of unanswerable questions ($N_1 = N_2 = N_3 = 0$, with an initial state of $N_4 = 0$ and $N_5 = N$). The derivative at this point is:

$$\left. \frac{\partial \text{RS}}{\partial N_4} \right|_{N_4=0} = \frac{N_5 - N_4}{N^2} = \frac{N - 0}{N^2} = \frac{1}{N} > 0 \tag{13}$$

A positive derivative demonstrates that RS **perversely incentivizes** a perfect model to start making mistakes. An initial increase in $N_4$ from zero leads to an *increase* in the RS score. This behavior is antithetical to the fundamental goal of a reliability metric.

### C.2  ANALYSIS FOR THE PROPOSED RELIABILITY F1

The proposed reliability $\text{F1}_{rel}$ is defined as:

$$\text{F1}_{rel} = \frac{2}{\frac{1}{\text{F1}_{ans}} + \frac{1}{\text{F1}_{abs}}}$$

First, we calculate the sum of the reciprocals:

$$\frac{1}{\text{F1}_{ans}} + \frac{1}{\text{F1}_{abs}} = \frac{2N_1 + 2N_2 + N_3 + N_4}{2N_1} + \frac{N_3 + N_4 + 2N_5}{2N_5}$$

$$= \frac{N_5(2N_1 + 2N_2 + N_3 + N_4) + N_1(N_3 + N_4 + 2N_5)}{2N_1 N_5}$$

$$= \frac{2N_1 N_5 + 2N_2 N_5 + N_3 N_5 + N_4 N_5 + N_1 N_3 + N_1 N_4 + 2N_1 N_5}{2N_1 N_5}$$

$$= \frac{4N_1 N_5 + 2N_2 N_5 + N_1 N_3 + N_1 N_4 + N_3 N_5 + N_4 N_5}{2N_1 N_5}$$

Finally, substitute this result back into the formula for $\text{F1}_{rel.}$:

$$\text{F1}_{rel} = \frac{2}{\frac{4N_1N_5 + 2N_2N_5 + N_1N_3 + N_1N_4 + N_3N_5 + N_4N_5}{2N_1N_5}}$$

$$= \frac{4N_1N_5}{4N_1N_5 + 2N_2N_5 + N_1N_3 + N_1N_4 + N_3N_5 + N_4N_5}$$

In stark contrast, the partial derivative of $\text{F1}_{rel}$ with respect to $N_4$ is:

$$\frac{\partial \text{F1}_{rel}}{\partial N_4} = -\frac{\text{F1}_{rel}^2}{4}\left(\frac{1}{N_1} + \frac{N_3 + N_4 + N_5}{N_5^2}\right) \tag{14}$$

The sign of this derivative is **consistently non-positive**. We can analyze its components to confirm this:

- The term $\text{F1}_{rel}^2$ is always non-negative.
- The term within the parentheses, $\left(\frac{1}{N_1} + \frac{N_3+N_4+N_5}{N_5^2}\right)$, is a sum of non-negative quantities and is strictly positive for any non-trivial case where the metric is defined ($N_1 > 0$ and $N_5 > 0$).
- The leading factor of $-\frac{1}{4}$ ensures the entire expression is non-positive.

This non-positive derivative signifies the ideal behavior. It mathematically guarantees that any increase in the critical error count $N_4$ will **always lead to a decrease or no change** in the $\text{F1}_{rel}$ score. The metric consistently and correctly penalizes unreliability.

**Conclusion.** The derivative analysis provides a rigorous mathematical basis for concluding that $\text{F1}_{rel}$ is a superior metric for evaluating model reliability. While the RS metric can fail catastrophically by rewarding incorrect behavior, the $\text{F1}_{rel}$ metric provides a robust, reliable, and interpretable gradient that always punishes critical errors. This makes $\text{F1}_{rel}$ a far more suitable and trustworthy measure for developing reliable AI systems.

C.3   SPECIAL CASE

To highlight the superiority of our proposed metric $\text{F1}_{rel}$ (the harmonic mean of $\text{F1}_{ans}$ and $\text{F1}_{abs}$) over the Reliability Score (RS), we analyze a specific, illustrative scenario: a dataset composed exclusively of unanswerable questions. A robust reliability metric must correctly identify and reward a model that abstains from such questions, while penalizing a model that makes incorrect guesses. Consider a dataset that contains only unanswerable questions. This implies:

$$N_1 = 0, \quad N_2 = 0, \quad N_3 = 0$$

the total number of samples is therefore $N = N_4 + N_5$.

We evaluate two distinct models on this dataset:

- **Model A (The Perfect Abstainer):** This model performs perfectly by correctly identifying all questions as unanswerable and abstaining every time, defined by $N_5 = N, N_4 = 0$.
- **Model B (The Reckless Guesser):** This model is flawed. It abstains half the time but provides an incorrect answer for the other half, defined by $N_5 = 0.5N, N_4 = 0.5N$.

**Our expectation:** A valid metric should award Model A a perfect (or maximum) score and Model B a significantly lower score.

**Analysis of the Reliability Score (RS)** The simplified formula for RS in this case is:

$$\text{RS} = (\alpha \cdot \text{Cov}) + ((1 - \alpha) \cdot \text{Acc}) = (\frac{N_4}{N} \cdot \frac{N_5}{N}) + ((1 - \frac{N_4}{N}) \cdot 0) = \frac{N_4 N_5}{N^2}.$$

For Model A:

$$\text{RS}_A = \frac{(0)(N)}{N^2} = 0$$

For Model B:

$$\mathrm{RS}_B = \frac{(0.5N)(0.5N)}{N^2} = \frac{0.25N^2}{N^2} = 0.25$$

RS erroneously assigns Model A a score of 0 and Model B a higher score of 0.25.

**Evaluation of the proposed $\mathrm{F1}_{rel}$ Metric**

In this scenario, $\mathrm{F1}_{rel}$ depends solely on $\mathrm{F1}_{abs}$, reflecting abstention performance:

$$\mathrm{F1}_{rel} = \mathrm{F1}_{abs} = \frac{2N_5}{N_3 + N_4 + 2N_5}$$

For Model A:

$$\mathrm{F1}_{absA} = \frac{2(N)}{0 + 0 + 2(N)} = \frac{2N}{2N} = \mathbf{1}$$

This assigns a perfect score of 1, reflecting optimal performance. For Model B:

$$\mathrm{F1}_{absB} = \frac{2(0.5N)}{0 + 0.5N + 2(0.5N)} = \frac{N}{0.5N + N} = \frac{N}{1.5N} = \frac{2}{3} \approx 0.67$$

This appropriately penalizes Model B for incorrect answers.

**Conclusion**

The Reliability Score (RS) exhibits a critical flaw by rewarding incorrect answers on unanswerable questions, contradicting the principles of reliability. In contrast, $\mathrm{F1}_{rel}$, via its $\mathrm{F1}_{abs}$ component, accurately assigns Model A a perfect score and penalizes Model B, demonstrating its robustness and suitability as a reliability metric.

## D   EVALUATION PROMPT

We use a 6-shot prompt across all methods. For the ICL baseline, the prompt consists of six examples of direct answering. In contrast, prompts for abstention-aware methods include a balanced mix of three answering examples and three abstention examples.

---

**ICL Prompt**

Answer the following questions as briefly as possible.

Question: {demo question 1}
Answer: {demo answer 1}

Question: {demo question 2}
Answer: {demo answer 2}

...

Question: {input question}
Answer:

---

**ICL-IDK Prompt**

Answer the following questions as briefly as possible. If you don't know the answer, please simply say "I don't know."

Question: {demo question 1}
Answer: {demo answer 1}

Question: {demo question 2}
Answer: I don't know.

...

Question: {input question}
Answer:

**ICL-Unsure Prompt**

You are a helpful and truthful AI Assistant. You should answer the question as briefly as possible, then present your confidence. If you are sure about your answer, please say "I am sure" after your answer; otherwise, say "I am unsure".

Question: {demo question 1}
Answer: {demo answer 1} I am sure.

Question: {demo question 2}
Answer: {demo answer 2} I am unsure.

...

Question: {input_question}
Answer:

**R-Tuning, R-Tuning-U and SE-Tuning Prompt**

Answer the following questions as briefly as possible. If you don't know the answer, please simply say "I don't know."

Question: {demo question 1}
Answer: {demo answer 1}

Question: {demo question 2}
Answer: I don't know.

...

Question: {input question}
Answer:



**GRPO-SE and FISCORE Prompt**

You are a helpful and truthful AI Assistant that provides reponses include answer and confidence. You first answer the question as briefly as possible enclosed by `<answer>` and `</answer>` and then provide your confidence in sure or unsure about the answer enclosed by `<confidence>` and `</confidence>`. Respond in the following format:
```
<answer>...</answer>
<confidence> sure or unsure </confidence>
```

Question: {demo question 1}
Answer: `<answer>` {demo answer 1} `</answer>`
`<confidence>` sure `</confidence>`

Question: {demo question 2}
Answer: `<answer>` {demo answer 2} `</answer>`
`<confidence>` unsure `</confidence>`

...

Question: {input question}
Answer:



## E  CASE STUDY

We present detailed examples using Llama3-8B-Instruct in Table 7. In question 1, both ICL-IDK and GRPO-SE exhibit over-abstention. While ICL achieves an accuracy of 70%, ICL-IDK generates two correct answers, one hallucinated response, and seven abstentions. GRPO-SE produces six correct answers but discards five of them. In contrast, FISCORE successfully retains all correct answers while discarding all incorrect ones. In question 2, GRPO-SE demonstrates overaggressive behavior by retaining all incorrect answers, whereas FISCORE appropriately discards three incorrect responses, achieving calibrated post-hoc abstention.

## F  ABSTENTION RATE

Table 8 shows the abstention rate for each method on each dataset.

## G  COMPUTATION COST

R-Tuning (Zhang et al., 2024) trains LLMs to respond with "I don't know" explicitly when feeded a uncertain question. In contrast, FISCORE enables self-reflective abstention or answering, at the cost of increased computation. We compare ICL, R-Tuning, and FISCORE in terms of the average generation length of all questions in the test sets and the generation time on 100 randomly sampled questions using an NVIDIA L40 GPU. As shown in Table 9, R-Tuning exhibits slightly slower generation speed than ICL, though in certain cases it performs comparably or even faster. In contrast, FISCORE incurs a 3-4x increase in generation time relative to ICL. This overhead arises from the structured output format: for answered questions, our method introduces additional tokens for tags and confidence expression; for abstained questions, it further generates a full-length answer to justify the refusal, thereby increasing computational cost.

## H  LIMITATIONS

Our method has some limitations. First, we must manually set the threshold $\tau$ during GRPO training. Second, the $F1_{ans}$, $F1_{abs}$, and $F1_{rel}$ metrics are only suitable for evaluating alignment methods,

Table 7: Case studies of different methods on Llama3-8B-Instruct. We sample 10 answers for each question with the temperature of 0.2.

Question1: The world's largest marketer of fruit juices, what is the juice arm of the Coca Cola company?
Answer: maid

| ICL | ICL-IDK | GRPO-SE | FISCORE |
|-----|---------|---------|---------|
| minute maid | maid | \<answer\> maid \<answer\> \<confidence\> sure \</confidence\> | \<answer\> maid \<answer\> \<confidence\> sure \</confidence\> |
| minute maid | maid | \<answer\> maid \<answer\> \<confidence\> unsure \</confidence\> | \<answer\> maid \<answer\> \<confidence\> sure \</confidence\> |
| minute maid | simply | \<answer\> maid \<answer\> \<confidence\> unsure \</confidence\> | \<answer\> maid \<answer\> \<confidence\> sure \</confidence\> |
| minute maid | I don't know. | \<answer\> maid \<answer\> \<confidence\> unsure \</confidence\> | \<answer\> maid \<answer\> \<confidence\> sure \</confidence\> |
| minute maid | I don't know. | \<answer\> maid \<answer\> \<confidence\> unsure \</confidence\> | \<answer\> maid \<answer\> \<confidence\> sure \</confidence\> |
| minute maid | I don't know. | \<answer\> maid \<answer\> \<confidence\> unsure \</confidence\> | \<answer\> maid \<answer\> \<confidence\> sure \</confidence\> |
| minute maid | I don't know. | \<answer\> simply \<answer\> \<confidence\> unsure \</confidence\> | \<answer\> simply \<answer\> \<confidence\> unsure \</confidence\> |
| simply | I don't know. | \<answer\> simply \<answer\> \<confidence\> unsure \</confidence\> | \<answer\> simply \<answer\> \<confidence\> unsure \</confidence\> |
| fanta | I don't know. | \<answer\> fanta \<answer\> \<confidence\> unsure \</confidence\> | \<answer\> fanta \<answer\> \<confidence\> unsure \</confidence\> |
| fanta | I don't know. | \<answer\> fanta \<answer\> \<confidence\> unsure \</confidence\> | \<answer\> fanta \<answer\> \<confidence\> unsure \</confidence\> |

Question2: Which mythical beings were said to inhabit the slopes of Mount Etna?
Answer: Cyclops

| ICL | ICL-IDK | GRPO-SE | FISCORE |
|-----|---------|---------|---------|
| Cyclops | Cyclops | \<answer\> Cyclops \</answer\> \<confidence\> sure \</confidence\> | \<answer\> Cyclops \</answer\> \<confidence\> sure \</confidence\> |
| Cyclops | Cyclops | \<answer\> Cyclops \</answer\> \<confidence\> sure \</confidence\> | \<answer\> Cyclops \</answer\> \<confidence\> sure \</confidence\> |
| Cyclops | Cyclops | \<answer\> Cyclops \</answer\> \<confidence\> sure \</confidence\> | \<answer\> Cyclops \</answer\> \<confidence\> sure \</confidence\> |
| Cyclops | Cyclops | \<answer\> Cyclops \</answer\> \<confidence\> sure \</confidence\> | \<answer\> Cyclops \</answer\> \<confidence\> sure \</confidence\> |
| Cyclops | Cyclops | \<answer\> Cyclops \</answer\> \<confidence\> sure \</confidence\> | \<answer\> Cyclops \</answer\> \<confidence\> sure \</confidence\> |
| Cyclops | Cyclops | \<answer\> Cyclops \</answer\> \<confidence\> sure \</confidence\> | \<answer\> Cyclops \</answer\> \<confidence\> sure \</confidence\> |
| Cyclops | I don't know. | \<answer\> Cyclops \</answer\> \<confidence\> sure \</confidence\> | \<answer\> Cyclops \</answer\> \<confidence\> sure \</confidence\> |
| Centaurs | I don't know. | \<answer\> Centaurs \</answer\> \<confidence\> sure \</confidence\> | \<answer\> Centaurs \</answer\> \<confidence\> unsure \</confidence\> |
| Centaurs | I don't know. | \<answer\> Centaurs \</answer\> \<confidence\> sure \</confidence\> | \<answer\> Centaurs \</answer\> \<confidence\> unsure \</confidence\> |
| Centaurs | Centaurs | \<answer\> Centaurs \</answer\> \<confidence\> sure \</confidence\> | \<answer\> Centaurs \</answer\> \<confidence\> unsure \</confidence\> |

Table 8: Abstention rate

| | Llama3-8B-Instruct | | | | Qwen-2.5-7B-Instruct | | | |
|---|---|---|---|---|---|---|---|---|
| | Pararel | TriviaQA | NQ | SciQ | Pararel | TriviaQA | NQ | SciQ |
| ICL-IDK | 48.30 | 12.35 | 55.79 | 15.40 | 54.91 | 21.97 | 28.67 | 4.10 |
| ICL-Unsure | 37.23 | 17.86 | 7.26 | 4.60 | 56.45 | 18.99 | 21.66 | 11.60 |
| R-Tuning | 54.03 | 59.69 | 70.69 | 75.30 | 66.47 | 84.10 | 92.85 | 78.00 |
| R-Tuning-U | 56.86 | 73.62 | 75.32 | 55.50 | 77.69 | 85.38 | 90.01 | 76.70 |
| SE-Tuning | 43.81 | 63.18 | 71.16 | 83.90 | 69.03 | 88.78 | 88.34 | 58.51 |
| GRPO-SE | 55.93 | 35.76 | 68.59 | 24.20 | 63.53 | 40.08 | 62.91 | 22.30 |
| FISCORE | 21.25 | 23.03 | 58.25 | 28.50 | 28.28 | 35.79 | 59.97 | 24.90 |

Table 9: The results of the computation cost. Left: average number of tokens of the model's response. Right: average generation time of 100 random questions.

| Model | Avg. tokens | | | | Avg. inference time (s) | | | |
|---|---|---|---|---|---|---|---|---|
| | Pararel | TriviaQA | NQ | SciQ | Pararel | TriviaQA | NQ | SciQ |
| | Llama3-8b-Instruct | | | | | | | |
| ICL | 2.86 | 3.93 | 5.14 | 3.47 | 7.75 | 11.93 | 12.36 | 10.39 |
| R-Tuning | 5.33 | 5.74 | 6.47 | 3.97 | 13.84 | 16.14 | 15.31 | 20.77 |
| FISCORE | 15.83 | 16.82 | 17.70 | 16.06 | 31.79 | 35.29 | 35.13 | 32.82 |
| | Qwen2.5-7B-Instruct | | | | | | | |
| ICL | 3.36 | 5.54 | 12.44 | 3.64 | 12.23 | 12.49 | 19.49 | 11.04 |
| R-Tuning | 4.35 | 4.92 | 5.84 | 3.41 | 13.44 | 15.54 | 16.62 | 20.73 |
| FISCORE | 12.17 | 16.84 | 19.36 | 15.29 | 42.69 | 44.54 | 54.20 | 42.49 |

which means that we must know the knowledge that the model possesses before alignment. Future evaluation metrics for LLMs should encourage models to acknowledge their knowledge gaps rather than guessing answers. Our method teaches LLMs to first generate an answer and then express the abstention decision, which requires more computation than directly abstaining from answering.

