# OpenReview forum: "Teaching LLMs to Abstain via Fine-Grained Semantic Confidence Reward"
_ICLR.cc/2026/Conference — Submitted to ICLR 2026_

### Official Review · Reviewer_otR2 · 2025-10-21

**Soundness:** 3
**Presentation:** 3
**Contribution:** 3
**Rating:** 4
**Confidence:** 4

**Summary:**

Authors present a novel method for teaching LLMs to abstain from answering queries when they are uncertain. Their method is based on training an LLM via the following process:

1) for a given query, sample multiple responses
2) cluster these responses
3) assign "high/low confidence" labels to said clusters based on the cluster's size
4) the model is rewarded if its self-estimated high/low confidence aligns with the cluster-based label

Their method shows strong generalization to OOD data, which baselines do not.

Authors additionally introduce a novel reliability metric that combines helpfulness and truthfulness together.

**Strengths:**

The paper's writing is clear, the method and novel metric are explained effectively.

The metric seems to be effectively justified (Appendix C.1).

The OOD generalization is very compelling.

Ablations are well-done.

**Weaknesses:**

I'm not sure that the method is motivated effectively enough (that we want a per-sample abstention boundary not a global one per-query). I understand the concept, but what exactly happens e.g. at test time? The clustering step etc. does not happen, correct? So what are we really gaining when compared to existing methods that use binary labels based on correct/incorrect answers or a single entropy-based reward per question?

Can we not just use this method of clustering at inference time, no training needed?

How exactly is your signal better than competing signals? This feels hand-wavy. What is the exact mechanism? It's very unclear to me.

Are there some confounding variables here w.r.t. training methods? Some competing methods are pure supervised fine-tuning vs. others using RL.

The distinction between GRPO-SE and FISCORE is very hazy to me. GRPO-SE already does clustering, no? So it's multiple rewards per question (one for each response) vs. 1 aggregated reward per question, is essentially what we're comparing? Is it the clustering specifically that's helping, or is it the fine-grained per-sample rewards/denser/richer reward signal that's helping?

**Questions:**

Why don't you try just using continuous rewards based on the cluster size proportion vs. the total, vs. a hard boundary?

Doing a causal intervention at inference time would really strengthen the mechanistic explanation of what's going on here. Manipulate the answer and measure confidence change:

1. Generate answer A from model
2. Manually replace A with answer B (from a different cluster during training)
3. Force model to generate confidence for B
4. Does confidence appropriately change based on what cluster B would belong to?

This seems like it would effectively test if the model internalized a representation of the clusters/of its own internal answer distribution frequency.

---

> ### Author Response · Authors · 2025-11-21
> **Author Response to Reviewer otR2**
>
> ### Thank you for the valuable review! We provide detailed responses to your points below.
>
>
> > **Response to the concerns about the motivation, the specific mechanism, and the difference between our method (FISCORE) and baselines like GRPO-SE.**
>
> Thank you for raising this issue. Let us clarify our contributions. The GRPO-SE baseline employs the same reward policy for all samples, whereas our approach utilizes sample-specific rewards. For example: suppose the model samples 10 times for `What is the capital of France?` After semantic clustering, 8 are `Paris` (large cluster, high confidence), 2 are `London` (small cluster, low confidence) [see Figure 2(b) in the manuscript for a similar example]. Global Entropy (GRPO-SE) computes one low global entropy (high consensus) and gives all 10 samples the same positive reward if the model outputs sure (for being "low entropy"). This incorrectly rewards the `London` samples. FISCORE (Ours) checks individually: For the 8 `Paris` samples: If the model says sure, it will get a positive reward. For the two `London` samples: If the model says sure, it will get a negative reward (penalty). Conclusion: Our fine-grained semantic confidence reward teaches the model to express its confidence at a fine-grained level, enabling it to make abstention decisions more aligned with its internal knowledge boundaries.
>
> > **Regarding the test-time procedure and why we do not simply use clustering during inference**
>
> Thank you for raising this question. Clustering at inference time is feasible but it is computationally too expensive. It requires sampling the model many times ($G$ times) and running a large NLI model to compare them. This would make the system very slow for real-time use. Instead, we use the confidence signal to teach the model to learn to abstain appropriately. During training, the model learns to express its confidence about each answer individually. Therefore, at test time, we do not need to sample or cluster. We simply perform a single, fast greedy search. The model outputs its answer and a calibrated confidence tag (sure or unsure), where we treat unsure as an abstention decision.
>
> > **Are there some confounding variables here w.r.t. training methods...**
>
> Thank you for raising this question. We address this with a controlled comparison. As detailed in Section 4.2, FISCORE and GRPO-SE use the exact same RL algorithm. The only difference between them is the confidence reward function. We included some SFT baselines to validate the necessity of reinforcement learning, as it demonstrates better generalization capabilities.
>
>
> >**Response to Q1: Why not use continuous rewards?**
>
> Thank you for raising this question. Since we focus on the problem of LLM abstention—where the model must make a binary choice to either answer or refuse—both our method and the baselines employ a binary discrete reward. If continuous rewards were used for training, a threshold would need to be manually set after training to determine whether the model should abstain. Furthermore, continuous rewards make training more complex, requiring carefully designed rewards and significantly more data.
>
> > **Response to Q2: Regarding the causal intervention.**
>
> Thank you for your suggestion regarding causal intervention at inference time. However, in our setting such manual manipulation is not necessary, because we can directly observe multiple candidate answers together with their confidence scores. As shown in Appendix E of the manuscript, we sample 10 responses for each method. In our approach, answers that appear with higher frequency are consistently labeled with sure confidence, whereas less frequent answers are labeled as unsure. This indicates that the model has indeed learned to express its internal confidence distribution via our method. If we have misunderstood your intended point, we would be glad to clarify further.

---

> > ### Comment · Reviewer_otR2 · 2025-11-23
> >
> > I thank the authors for their responses, they have clarified some things to me.
> >
> > The threshold for high/low confidence cannot be modified at runtime then, right? Every different threshold would need retraining from scratch. Continuous rewards would allow you to be able to threshold dynamically at runtime based on the use case, no?
> >
> > I appreciate the direction to Appendix E, it missed my eye on first pass. Compelling evidence.
> >
> > So if if were to sum up your approach, it is essentially a matter of avoiding the multiple rollouts that would be required to apply the semantic entropy + clustering method at runtime. Hence transforming it into a reward to be trained with RL.

---

> > > ### Author Response · Authors · 2025-11-24
> > > **Author Response to Reviewer otR2**
> > >
> > > We sincerely thank you for acknowledging the clarity and evidence provided in our response.
> > >
> > > We also appreciate your insightful response about the continuous rewards. As you mentioned, continuous rewards can dynamically adjust thresholds after inference, thereby avoiding retraining on different datasets. Following your suggestion, we trained Qwen2.5-7B-Instruct using continuous rewards. Specifically, during training, the model samples 10 answers and output corresponding confidence scores ranging from 10% to 100%. The reference confidence value is the ratio of the answer's cluster size to 10. We employed negative mean squared error as the confidence reward function for training, with all other settings identical to those for discrete rewards. The results are shown in the table below, where both discrete and continuous rewards results are obtained from model trained on parallel datasets. For each dataset, we select the best threshold to achieve the highest $\text{F1} _ {rel}$ as the results for continuous rewards. It is noticeable that continuous rewards consistently outperform discrete rewards across various datasets. Furthermore, continuous rewards adapt to different datasets without requiring retraining, whereas discrete rewards require retraining for different scenarios. In practical applications, training with continuous rewards across multiple datasets can produce better performance.
> > >
> > >
> > > #### Table 1: Comparison of Discrete and Continuous Rewards
> > > | Method | Pararel | | | | TriviaQA | | | | NQ | | | | SciQ | | | |
> > > |-----------|-------|-----|------|------|------|------|------|------|-----|-----|-----|-----|------|------|------|------|
> > > | | $\text{F1} _ {ans}$ | $\text{F1} _ {abs}$ | $\text{F1} _ {rel}$ | Acc | $\text{F1} _ {ans}$ | $\text{F1} _ {abs}$ | $\text{F1} _ {rel}$ | Acc | $\text{F1} _ {ans}$ | $\text{F1} _ {abs}$ | $\text{F1} _ {rel}$ | Acc | $\text{F1} _ {ans}$ | $\text{F1} _ {abs}$ | $\text{F1} _ {rel}$ | Acc |
> > > | Ours-discrete rewards | 80.8 | 72.9 | 76.6 | 51.9 | 83.3 | 79.1 | 81.1 | 51.3 | 57.3 | 80.1 | 66.8 | 20.4 | 79.2 | 54.5 | 64.6 | 58.0 |
> > > | Ours-continuous rewards | 74.9 | 82.2 | 78.4 | 33.3 | 83.0 | 82.9 | 82.9 | 49.1 | 58.1 | 85.0 | 69.0 | 17.8 | 79.4 | 56.8 | 66.2 | 60.8 |
> > >
> > >
> > > Your summary of our approach is accurate, and our method, GRPO-SE, and SE-Tuning are all based on this underlying principle.
> > >
> > >
> > > Thanks again for your feedback. If there are any concerns remaining, please let us know.

---

### Official Review · Reviewer_4ymx · 2025-10-30

**Soundness:** 3
**Presentation:** 3
**Contribution:** 2
**Rating:** 6
**Confidence:** 4

**Summary:**

This paper presents a novel and well-executed approach to a critical problem in LLM reliability: teaching models to abstain from answering questions beyond their knowledge scope. The proposed method, FISCORE, leverages reinforcement learning with a fine-grained, per-sample confidence reward derived from semantic clustering. It also analyzes the shortcomings of existing metrics and proposes a new metric, F1_rel, to capture the trade-off between helpfulness and truthfulness.

**Strengths:**

The fine-grained reward moves beyond coarse, aggregated uncertainty metrics, further optimizing the model's abstention behavior. The F1_rel proposed by the authors reflects a pursuit of balance between helpfulness and truthfulness, and their discussion on the effectiveness of F1_rel compared to existing metrics also provides valuable insight. The experimental section is thorough and well-designed, comparing the method to strong baselines and verifying its effectiveness, especially its generalizability. Besides, the paper is very well-written with a clear and logical structure.

**Weaknesses:**

1. The effectiveness and necessity of the confidence reward are questionable. (a) Effectiveness: The confidence reward essentially guides the model to be more confident and consistent in its output. This might incorrectly encourage hallucinations. For instance, given a complex question the model cannot answer, it might obtain multiple different incorrect answers across samples. The confidence reward could end up rewarding the most frequent of these incorrect outputs, thereby causing the model to be even more stubborn in its hallucinations. This phenomenon of potential reward hacking from such a confidence reward has already been discussed in [1]. (b) Necessity: Judging from the experimental results in Section 4.3, Figure 3(a), the necessity of the confidence reward seems weak. Even when the accuracy reward weight is increased to 4, at which point the confidence reward's weight is comparatively low and the overall reward largely degenerates into one focused only on accuracy, the model still achieves a high F_rel. Does this imply that the performance improvement stems from training with the accuracy reward rather than the confidence reward? Perhaps merely using the accuracy reward while prompting the model to generate a sure/unsure label (without factoring it into the reward) could also yield a model with a decent F_rel.
2. Limitations of the evaluation metric: The F1_rel metric favors a balanced model (where F1_ans and F1_abs are optimized simultaneously and are relatively close), as indicated by the data in Table 1. From the response distribution in Figure 4, one can also see that FISCORE actually generates more incorrect answers (N2+N4), implying poorer reliability. Concurrently, SE-Tuning generates fewer incorrect answers but performs more unnecessary abstentions (N3), which brings higher truthfulness and lower helpfulness, manifesting as a lower overall F1_rel. Thus, it seems the difference in F1_rel between methods depends more on the model's answering rate, that is, its trade-off preference between truthfulness and helpfulness. A model preferring balance is more likely to obtain a high F1_rel. This is inflexible. Many low-error-tolerance scenarios might prefer a model like R-Tuning, which sacrifices helpfulness but makes very few errors. Perhaps a hyperparameter could be introduced to adjust the weights of F1_ans and F1_abs to evaluate performance in different scenarios?
3. Insufficient ablation studies, including: (1) The use of a fixed abstention threshold of G/2: Intuitively, the value of the abstention threshold should be correlated with task difficulty. For a more difficult task, perhaps a lower threshold is required? Has the impact of the threshold's value on training been investigated? (2) Performance and impact of the semantic model: What is the performance of the currently used un-trained DeBERTa? Are there instances of inaccurate clustering? Perhaps some training would achieve better performance?

If my questions are resolved, I will consider raising the score.

References

[1] https://arxiv.org/abs/2505.21444

**Questions:**

1. Could you show a comparison of the rejection rates for different baselines on each dataset?
2. What is the importance of the format reward? I noticed you set a relatively high weight for the format reward (w_f=2) but provided no explanation. Was this the best value found in your experiments? Does the format reward have a significant impact on training?
3. Suggestion: The three weight values for R_total in equation (9) are important pieces of information. Consider stating them near equation (9) or in the implementation details. It took me some time to find their corresponding values in the appendix.

---

> ### Author Response · Authors · 2025-11-21
> **Author Response to Reviewer 4ymx (Part 1)**
>
> ### Thank you for your hard work and helpful feedback. Here are our responses to the review.
>
> >**Regarding the effectiveness of the confidence reward.**
>
> Thank you for raising this important point. Self-rewarded training (SWT)[1] may cause model collapse because it consistently encourages models to output majority-vote answers, which are not equivalent to correct answers. Although this method is effective in the early stages of training, it may later cause the model to discover that outputting incorrect answers and miscalibrating their confidence scores to extremely high values gets rewarded, ultimately leading to model collapse. However, we do not reward models for outputting majority-vote answers; instead, we reward models for outputting factually correct answers. The proposed semantic confidence rewards only calibrate the confidence expression after answers, ensuring the model's confidence level aligns with the semantic clustering results. We design a combined mechanism of accuracy rewards and confidence rewards, with accuracy rewards given higher weight to prevent the model from discarding its high-confidence answers and instead outputting uncertain ones. Our ablation experiments confirmed that removing the accuracy reward causes model collapse. This demonstrates that carefully designing the ratio between accuracy and confidence rewards can effectively prevent reward hacking. However, such a self-consistency-based approach cannot prevent models from outputting incorrect knowledge with high certainty prior to fine-tuning. Such hallucinations cannot be resolved through confidence or uncertainty[2] and require methods like SFT or knowledge editing.
>
> > **Regarding the necessity of the confidence reward.**
>
> Thank you for raising this important point. We conduct the ablation on our confidence reward. Experimental results indicate that removing the confidence reward significantly reduces the model's abstention capability. However, we observe that the model still rejects a small number of questions, demonstrating a basic rejection awareness. Although the confidence reward is given a small weight, it can enhance the model's ability to self-reflect on its own confidence.
>
> #### Table 1: Ablation on confidence reward function
> | | $\text{F1} _ {ans}$ | $\text{F1} _ {abs}$ | $\text{F1} _ {rel}$ | Acc |
> | :------|---- | ----|---- | ---- |
> | Ours| 83.3 | 79.1|81.1 | 51.3 |
> w/o confidence reward | 69.5 | 2.1 | 4.1 | 56.1|
>
>
>
> > **Limitations of the evaluation metric.**
>
> Thank you for pointing out this issue. $\text{F1} _ {rel}$ is indeed a trade-off, which primarily focuses on general scenarios. In low-tolerance scenarios, LLMs should place greater emphasis on abstention performance. We fully agree with your perspective, so we are introducing a hyperparameter for $\text{F1} _ {rel}$ to control the weighting between answering and abstention: $\text{F}\beta _ {rel}$ score:
> $$
>         \text{F} \beta_{{{rel}}} = (1 + \beta^2) \cdot \frac{\text{F1} _ {{\text{ans}}} \cdot \text{F1} _ {{\text{abs}}}}{\beta^2 \cdot \text{F1} _ {{\text{ans}}} + \text{F1} _ {{\text{abs}}}},
> $$
> when $\beta =1$, the metric is equivalent to our original $\text{F1} _ {rel}$. When $\beta=0$, the metric is equivalent to $\text{F1} _ {ans}$. When $\beta$ approaches infinity, the metric is equivalent to $\text{F1} _ {abs}$.
>
> > **The ablation on the abstention threshold.**
>
> We conduct ablation experiments on threshold settings using qwen2.5-7b-it on TriviaQA, with $G$ set to 10. As shown in the table, when the threshold is set to 5, $\text{F1} _ {rel}$ reaches its maximum value. Acc decreases as the threshold increases, while $\text{F1} _ {abs}$ exhibits the opposite trend. $\text{F1} _ {ans}$ increase initially before declining. This may occur when the threshold is set too low, causing the model to generate excessive hallucinations. The resulting harm from these hallucinations outweighs the benefits gained from correct answers. For low-tolerance scenarios, it is best to use a larger threshold.
>
> #### Table2: The ablation results on the abstention threshold.
>  | Threshold | $\text{F1} _ {ans}$ | $\text{F1} _ {abs}$ | $\text{F1} _ {rel}$ | Acc  |
> |------|----------|----------|----------|------|
> | 3    | 79.3     | 65.6     | 71.8     | 54.7 |
> | 4    | 82.4     | 75.4     | 78.8     | 53.0 |
> | 5    | 83.3     | 79.1     | 81.1     | 51.3 |
> | 6    | 82.1     | 78.3     | 80.1     | 49.9 |
> | 7    | 74.7     | 81.2     | 77.8     | 36.5 |
>
>
> ### Reference
> [1][Can Large Reasoning Models Self-Train?](https://arxiv.org/abs/2505.21444)
>
> [2][Trust Me, I’m Wrong: LLMs Hallucinate with Certainty Despite Knowing the Answer](https://aclanthology.org/2025.findings-emnlp.792/)
>
> Please see part 2 below.

---

> ### Author Response · Authors · 2025-11-21
> **Author Response to Reviewer 4ymx (Part 2)**
>
> > **Regarding the performance and impact of the semantic model.**
>
>
> The Deberta-v2-xlarge-mnli[3] is fine-tuned on the MNLI dataset, which includes 392K premise-hypothesis pairs. For a fair comparison, SE-Tuning, GRPO-SE and our method all use the Deberta-v2-xlarge-mnli. We selected 40 questions from Pararel and TriviaQA respectively, and for each question we sampled ten examples for manual verification. We found that Deberta-v2-xlarge-mnli achieved accuracies of 98\% and 96\% on these datasets. Fine-tuning the DeBERTa model on these manually annotated datasets will yield better results, but this process is labor-intensive. In addition, a small amount of one-way entailment answers may also impact performance. For example, for the question ``What field does Robert Bunsen work in?``, DeBERTa classifies the answers ``science`` and ``chemistry`` into two different categories. However, they have a hierarchical relationship and are not suitable for grouping together into the same cluster, or separating completely. Hierarchical clustering[4] and finer-grained semantic classification[5] can mitigate this issue.
>
> > **Q1: Comparison of the rejection rates for different baselines on each dataset.**
>
> As shown in the table, SFT methods such as SE-tuning exhibit extremely high rejection rates. Prompt-based methods demonstrate the lowest rejection rates. Reinforcement learning-based methods yield moderate rejection rates.
>
> | Method      | Pararel | TriviaQA | NQ    | SciQ  | Pararel | TriviaQA | NQ    | SciQ  |
> |-------------|---------|----------|-------|-------|---------|----------|-------|-------|
> |             | **Llama**             |             |             |             | **Qwen**             |             |             |             |
> | ICL-IDK     | 48.3    | 12.4     | 55.8  | 15.4  | 54.9    | 22.0     | 28.7  | 4.1   |
> | ICL-Unsure  | 37.2    | 17.9     | 7.3   | 4.6   | 56.5    | 19.0     | 21.7  | 11.6  |
> | R-Tuning    | 54.0    | 59.7     | 70.7  | 75.3  | 66.5    | 84.1     | 92.9  | 78.0  |
> | R-Tuning-U  | 56.9    | 73.6     | 75.3  | 55.5  | 77.7    | 85.4     | 90.0  | 76.7  |
> | SE-Tuning   | 43.8    | 63.2     | 71.2  | 83.9  | 69.0    | 88.8     | 88.3  | 58.5  |
> | GRPO-SE     | 55.9    | 35.8     | 68.6  | 24.2  | 63.5    | 40.1     | 62.9  | 22.3  |
> | FISCORE     | 21.3    | 23.0     | 58.3  | 28.5  | 28.3    | 35.8     | 60.0  | 24.9  |
>
>
>
> > **Q2: Regarding the format reward.**
>
> We observed that when $w_f:w_c$ is set to 1 or lower, sometimes the answer content is not fully enclosed within `<answer> </answer>` tags. This significantly impacts the accuracy of semantic clustering. Therefore, we set $w_f$ to twice the value of $w_c$, while $w_a$ is set to four times $w_c$, exceeding the sum of the $w_c$ and $w_f$. We aim to anchor on correctness, ensure correct formatting to guarantee semantic clustering accuracy, and only then establish an accurate semantic confidence reward that prevents reward hacking.
>
> > **Q3: Suggestion: Consider stating the reward weights near equation (9) or in the implementation details.**
>
> Thank you for this suggestion. We have added the reward weights setting into the implementation details for clarity in the new version.
>
> ### Reference
>
> [3]https://github.com/microsoft/DeBERTa
>
> [4] [SINdex: Semantic INconsistency Index for Hallucination Detection in LLMs](https://arxiv.org/abs/2503.05980)
>
> [5] [Semantic Density: Uncertainty Quantification for Large Language Models through Confidence Measurement in Semantic Space](https://arxiv.org/abs/2405.13845)

---

### Official Review · Reviewer_Jezb · 2025-11-01

**Soundness:** 3
**Presentation:** 3
**Contribution:** 2
**Rating:** 4
**Confidence:** 5

**Summary:**

This paper proposes a reinforcement learning (RL) framework, FISCoR (Fine-grained Semantic Confidence Reward), designed to mitigate LLM hallucinations by teaching the model to abstain from providing specific, incorrect factual claims within an otherwise correct response. The core innovation lies in replacing traditional coarse-grained reward signals (like overall answer correctness or sampling entropy) with a reward based on explicit, fine-grained semantic confidence tags generated by the model itself (e.g., tagging individual clauses as "sure" or "unsure"). This approach aims to align the model's internal knowledge boundaries with its external generation behavior more precisely than previous methods.

**Strengths:**

1. The primary strength is moving from sentence- or response-level uncertainty to semantic chunk-level confidence. This allows the model to differentiate between known and unknown information within a single output, promoting more nuanced abstention

2. The method operationalizes the metacognitive ability (self-assessed confidence) into a quantifiable reward signal for RL, offering a direct path to behavioral modification.

**Weaknesses:**

1. The approach relies on the model simultaneously generating the answer and its confidence. As evidenced by recent work on metacognitive decoupling (e.g., the Answer-Free Confidence Estimation (AFCE) framework[1]), eliciting the answer and confidence simultaneously can introduce a strong cognitive bias, leading to overconfidence. If the confidence signal itself is biased, the resulting RL reward (and thus the trained policy) will be flawed.

2. The "fine-grained" semantic confidence reward is fundamentally a discrete, binary, or limited-scale estimation (e.g., "sure" vs. "unsure"). This is a coarse quantization of the true latent uncertainty (which is better represented by continuous scores or sampling entropy/divergence, akin to ROC-AUC metrics[2] applied to model output probabilities). This discretization may discard valuable information and introduce unnecessary bias.

3. The paper proposes an integrated metric that equally balances helpfulness (answering) and truthfulness/reliability (abstention). This fixed, symmetric trade-off makes the metric difficult to interpret or optimize for real-world scenarios where the cost of error vs. the value of a correct answer is highly asymmetric (e.g., a search engine favors helpfulness, while a medical diagnosis system favors reliability). The metric should allow for scenario-specific weighting.

**Questions:**

1. Given that simultaneous elicitation of answers and confidence is known to induce overconfidence bias (as argued in prior work on decoupling cognition from metacognition), what experiments were conducted to verify the calibration of the fine-grained confidence tags before using them as the reward signal? Did you consider a decoupled, two-stage prompting method (like AFCE) to generate a less biased confidence reward?

2. The paper critiques coarse-grained signals but utilizes a discrete (quantized) confidence label. What is the performance trade-off between this discrete semantic label and existing continuous, latent uncertainty measures (e.g., sampling consistency, variance, or token log-probabilities) that have been shown to correlate strongly with correctness ?

3. The methodology of generating a binary tag (sure/unsure or similar) alongside the answer has precedents in instruction-tuning for abstention[3]. Please include in the paper a discussion and comparison with related work that uses simple sure/unsure tags for finetuning.

[1]https://arxiv.org/pdf/2506.00582

[2]https://arxiv.org/pdf/2305.14613

[3]https://arxiv.org/pdf/2503.02233

---

> ### Author Response · Authors · 2025-11-21
> **Author Response to Reviewer Jezb**
>
> ### Thank you for your constructive review. We provide detailed responses to your concerns below.
>
> > **Given that simultaneous elicitation of answers and confidence is known to induce overconfidence bias (as argued in prior work on decoupling cognition from metacognition), what experiments were conducted to verify the calibration of the fine-grained confidence tags before using them as the reward signal? Did you consider a decoupled, two-stage prompting method (like AFCE) to generate a less biased confidence reward?**
>
> Thanks for raising this valuable question. The two-stage prompting is an excellent improvement. We considered the issue of mutual influence between answers and confidence, so we attempted to adjust the ratio between accuracy reward and confidence reward. We found that setting a higher weight for accuracy reward generally resulted in higher overall model reliability. Experimental results can be found in Section 4.3 of the paper. Prior to this, we had not explored a two-stage prompting approach. To our knowledge, AFCE[1] is the first to decouple answer and confidence. Following your suggestion, we test the two-stage prompting method on initial LLMs. Within existing training frameworks, our method cannot achieve fine-tuning based on two-stage prompts. The method of combining two-stage prompts with fine-tuning is highly worthy of exploration, which we leave for future work.
>
> | Method | Pararel | TriviaQA | NQ | SciQ | Avg |
> |--------|---------|----------|----|---- |-----|
> | Llama3-8B-Instruct | |  |  |  |  |
> | IDK-Unsure | 69.3 | 55.0 | 25.9 | 28.0 | 44.6 |
> | IDK-Unsure-2stage prompt | 43.0 | 35.1 | 31.8 | 41.3 | 37.8 |
> | Qwen2.5-7B-Instruct | | | |  |  |
> | IDK-Unsure | 80.2 | 59.9 | 39.9 | 49.2 | 57.3 |
> | IDK-Unsure-2stage prompt | 64.0 | 65.9 | 54.7 | 59.5 | 61.0 |
>
>
> >**The paper critiques coarse-grained signals but utilizes a discrete (quantized) confidence label. What is the performance trade-off between this discrete semantic label and existing continuous, latent uncertainty measures (e.g., sampling consistency, variance, or token log-probabilities) that have been shown to correlate strongly with correctness ?**
>
> Our approach focuses on the issue of LLM abstention, where the model must decide whether to refuse to answer. Therefore, both our method and the baselines use discrete confidence labels. SFT methods such as R-Tuning, R-Tuning-U, and SE-Tuning all employ binary labels. GRPO-SE utilizes binary discrete rewards. To ensure a fair comparison, we did not directly contrast continuous rewards with the baseline. Continuous measures are typically used for uncertainty quantification or hallucination detection, where performance is generally evaluated based on the correlation between the output score and correctness—specifically, the Area under the Receiver Operating Characteristic Curve (AUROC) metric. Consequently, our method is not suitable for comparison with the approaches you mentioned. We conducted ablation experiments comparing continuous rewards with discrete rewards on the proposed method. The results, presented in Table 5 of the updated manuscript, demonstrate that continuous rewards yield superior performance when compared to the optimal threshold and discrete rewards. If we have misunderstood your point, please correct us.
>
>
>
>
> >**Regarding the limitations of the reliability metric.**
>
>
> Thank you for pointing out this issue. To make the reliability metric applicable across various scenarios, we add a weighting parameters into $\text{F1} _ {rel}$:
>
> $$
>         \text{F} \beta_{{{rel}}} = (1 + \beta^2) \cdot \frac{\text{F1} _ {{\text{ans}}} \cdot \text{F1} _ {{\text{abs}}}}{\beta^2 \cdot \text{F1} _ {{\text{ans}}} + \text{F1} _ {{\text{abs}}}},
> $$
> when $\beta = 1$, the metric is equivalent to our original $\text{F1} _ {rel}$. When $\beta=0$, the metric is equivalent to $\text{F1} _ {ans}$. When $\beta$ approaches infinity, the metric is equivalent to $\text{F1} _ {abs}$. In scenarios like search engines, a small $\beta$ value is appropriate, while in scenarios like healthcare and law, a large $\beta$ is more suitable.
>
>
> >**The methodology of generating a binary tag (sure/unsure or similar) alongside the answer has precedents in instruction-tuning for abstention. Please include in the paper a discussion and comparison with related work that uses simple sure/unsure tags for finetuning.**
>
> Thanks for the suggestion. We have added this to the related work section in the new version.
>
>
> ### Reference
> [1][Do Language Models Mirror Human Confidence? Exploring Psychological Insights to Address Overconfidence in LLMs](https://arxiv.org/abs/2506.00582)

---

### Official Review · Reviewer_DDMx · 2025-11-02

**Soundness:** 2
**Presentation:** 3
**Contribution:** 2
**Rating:** 4
**Confidence:** 3

**Summary:**

The paper employs a reinforcement learning approach to teach the model to abstain. Answers are first clustered based on semantic similarity, and the model is deemed confident enough to respond (receiving a reward of 1) only when a cluster reaches a sufficient size. Experiments demonstrate the effectiveness of this method.

**Strengths:**

* The motivation is clear and the problem is significant.
* The writing is clear and I can understand this work.

**Weaknesses:**

* A key baseline is missing: TTRL [1], a reward design based on majority vote. Essentially, the method proposed in this paper, clustering answers semantically and determining confidence based on the number of answers in each cluster, is fundamentally an extension of majority voting. In other words, answers that appear more frequently across repeated samples are assigned higher confidence.
* The exploration of reward combinations is insufficient. Some studies have shown that omitting the format reward may lead to better performance. Have you tried this approach?
* Compared to not considering abstention (i.e., $w_c=0$), how does the task performance (accuracy) change?

[1] https://arxiv.org/pdf/2504.16084

**Questions:**

* Is it appropriate to measure uncertainty directly using the likelihood of each answer?

---

> ### Author Response · Authors · 2025-11-21
> **Author Response to Reviewer DDMx**
>
> ### Thank you for the valuable review! We provide detailed responses to your points below.
>
> >  **A key baseline is missing: TTRL[1], a reward design based on majority vote...**
>
> Thanks for this suggestion. We acknowledge that our method and TTRL[1] share a similar core idea in leveraging the majority vote principle in self-training, and we will discuss TTRL in the main text of the manuscript. We have carefully read TTRL. We compared our method with TTRL and found that their objectives and reward mechanisms differ fundamentally. TTRL's objective is to enhance the model's overall accuracy in verifiable answers by leveraging majority vote signals. Our approach aims to explicitly align the model's expressed confidence with semantic confidence (${R}_{\text{c}}$) through voting results obtained via semantic clustering. This distinction is crucial: our approach targets the abstention problem by training the model to output "sure" for majority-vote (large cluster) answers and "unsure" for minority (small cluster, non-consensus) answers, thereby learning when to abstain. In contrast, TTRL's reward mechanism is not designed or optimized for this functionality.
>
> > **Ablation on format reward.**
>
>
> #### Table 1: Ablation on reward function
> | | $\text{F1}_{ans}$ | $\text{F1}_{abs}$ | $\text{F1}_{rel}$ | Acc |
> | :------|---- | ----|---- | ---- |
> | Ours| 83.3 | 79.1|81.1 | 51.3 |
> | w/o format reward | 71.5 | 43.6 | 54.2 | 51.5|
> | w/o confidence reward | 69.5 | 2.1 | 4.1 | 56.1|
>
> Thank you for this valuable comment. We conduct ablation studies on TriviaQA using Qwen2.5-7B-Instruct. As shown in Table 1, removing the format reward causes the model's accuracy to remain nearly unchanged, but significantly degrades its abstention performance. This is because during training, we extract answers using rule-based extraction. When the model's answer is not enclosed within `<answer>` and `</answer>`, the accuracy of semantic clustering decreases, ultimately affecting the decision of whether to abstain or not. Therefore, the format reward is crucial to our method in practice.
>
> > **Ablation on confidence reward.**
>
> Thank you for this valuable comment. We conduct an ablation study on the fine-grained semantic confidence reward. As shown in Table 1, removing the confidence reward improved the model's accuracy (Acc), but caused a significant drop in $\text{F1} _ {ans}$. Furthermore, the model's $\text{F1} _ {abs}$ is nearly zero, indicating that the model has almost failed to learn to reject low-confidence answers.
>
> > **Question: Is it appropriate to measure uncertainty directly using the likelihood of each answer?**
>
> Thanks for raising this point. Likelihood can certainly be used for uncertainty estimation. However, existing studies show that such logits-based (including likelihood, PPL, probability) approaches underperform compared to sampling-based methods[2][3]. Sampling-based techniques are widely adopted in abstention fine-tuning[4][5], and we follow this established practice.
>
> ### Reference
>
> [1][TTRL: Test-Time Reinforcement Learning](https://arxiv.org/abs/2504.16084)
>
>
> [2][Semantic Uncertainty: Linguistic Invariances for Uncertainty Estimation in Natural Language Generation](https://openreview.net/forum?id=VD-AYtP0dve)
>
> [3][ICR Probe: Tracking Hidden State Dynamics for Reliable Hallucination  Detection in LLMs](https://aclanthology.org/2025.acl-long.880/)
>
> [4][R-Tuning: Instructing Large Language Models to Say ‘I Don’t Know’](https://aclanthology.org/2024.naacl-long.394/)
>
> [5][Fine-Tuning Large Language Models to Appropriately Abstain with Semantic Entropy](https://arxiv.org/abs/2410.17234)

---

> > ### Comment · Reviewer_DDMx · 2025-11-26
> >
> > Thank you for your response. Most of my concerns have been addressed, so I have decided to raise my score to 6. However, I still feel that using a majority vote to estimate confidence is a rather straightforward and somewhat trivial idea. Since I am not an expert in LLM reliability, I have accordingly lowered my confidence in this assessment.

---

### Author Response · Authors · 2025-12-03
**Summary of responses**

# Summary of responses
We are thankful for the reviewers' thoughtful and constructive feedback. In response, we have addressed each point and revised the manuscript accordingly. Below is a summary.

## The motivation of our work
Our core idea is simple yet effective: when training LLMs, we sample multiple answers for the same question and perform clustering, discarding minor-clusters (low semantic confidence) answers while retaining major-cluster (high semantic confidence) answers, and rewarding at the sample granularity level. In contrast, the baseline rewards at the question granularity level—that is, it assigns the same reward to all answers—which can sometimes mislead the model. Figure 1 and Appendix E in the paper provide intuitive examples.

## Ablation studies
We supplemented Section 4.3 with ablation experiments on reward functions and abstention threshold. We demonstrate the effectiveness of the proposed confidence reward. And the majority-vote methods referenced by Reviewer DDMx and Reviewer 4ymx are essentially the same method. We explained that training solely with a majority-vote reward causes model collapse due to reward hacking (An intuitive example: the model gets rewards when it consistently answers 1+1=3). We avoided this by using a weighted combination of accuracy reward and confidence reward, as also shown in our ablation experiments.

## Continious confidence reward
Reviewers Jezb and otR2 suggested that continuous rewards would be preferable. We emphasized that, for fair comparison, our method followed the baseline by using discrete rewards. We also supplemented the experiments with continuous rewards.

## The reliability metric
Based on the suggestions from Reviewer Jezb and Reviewer 4ymx, we introduce a hyperparameter to the proposed F1 reliability metric to flexibly adapt to evaluations in scenarios with varying weights for helpfulness and truthfulness.

## Limitations
We manually verified the accuracy of the NLI model and found it exceeds 95%, with negligible impact on model performance. We acknowledge this as an unavoidable limitation. Reviewer Jezb noted that simultaneously eliciting answers and confidence scores introduces confidence bias. Both our method and the baseline have this problem, and we mitigated by adjusting the weighting between accuracy reward and confidence scores.

## Other details
We have updated other details in the revised edition.

---

### Meta-Review · Area_Chair_p7Ue · 2026-01-10

**Summary:**

This work proposes a reinforcement learning framework for improving LLM abstention behavior using a Fine-Grained Semantic Confidence Reward (FSCR). The method samples multiple candidate answers, clusters them semantically, and trains the model to retain answers from high-confidence clusters while discarding low-confidence ones, enabling more precise, sample-specific abstention than prior coarse uncertainty-based approaches. The authors also introduce a metric to evaluate abstention reliability.

Reviewers initially raised concerns about the missing comparison with simple majority-vote baselines. In addition, they questioned whether the use of discrete, cluster-based confidence rewards was justified compared to continuous alternatives. Further concerns were raised about the validity and flexibility of the proposed reliability metric.

During the rebuttal, the authors addressed most of these concerns, but one major issue remains: the lack of a proper comparison with continuous confidence rewards, as raised by both Reviewers Jezb and otR2. Although the authors argue that continuous rewards are not a fair comparison because the task focuses on binary abstention (answer or refuse), we believe that continuous rewards still constitute a strong and relevant baseline as long as they can support the binary decision, as suggested by the additional experiments. Therefore, we recommend rejecting this work.

**Reviewer Concerns:**

The authors have addressed most of the reviewers’ concerns, except for the major issue raised by both Reviewers Jezb and otR2 regarding the comparison with continuous rewards. The results show that continuous rewards achieve better performance, while the authors argue that they are not strongly relevant baselines because the task focuses on binary abstention (answer or refuse).

**Reviewer Scores:**

Reviewer DDMx raised their score from 4 to 6 after the rebuttal.

Reviewers Jezb and otR2 would maintain their scores at 4, as the issue regarding continuous rewards has not been adequately addressed.

Reviewer 4ymx would maintain their score at 6.

---

### Decision · Program_Chairs · 2026-01-26

Reject